# The extracellular matrix supports breast cancer cell growth under amino acid starvation by promoting tyrosine catabolism

Mona Nazemi[1], Bian Yanes[1☯], Montserrat Llanses Martinez[1,2☯], Heather J. Walker[3], Khoa Pham[3], Mark O. Collins[1,3], Frederic Bard[2,4], Elena Rainero[1]*

1 School of Biosciences, The University of Sheffield, Sheffield, United Kingdom, 2 Institute of Molecular and Cell Biology, Singapore, 3 biOMICS Facility, School of Biosciences, The University of Sheffield, Sheffield, United Kingdom, 4 Centre de Recherche en Cancerologie de Marseille, CRCM, Marseille, France

☯ These authors contributed equally to this work.
* e.rainero@sheffield.ac.uk

**Data Availability Statement:** All the numerical data are provided as excel files in the supplementary data files (S1 Data to S16 Data). The metabolomics

## Abstract

Breast tumours are embedded in a collagen I-rich extracellular matrix (ECM) network, where nutrients are scarce due to limited blood flow and elevated tumour growth. Metabolic adaptation is required for cancer cells to endure these conditions. Here, we demonstrated that the presence of ECM supported the growth of invasive breast cancer cells, but not non-transformed mammary epithelial cells, under amino acid starvation, through a mechanism that required macropinocytosis-dependent ECM uptake. Importantly, we showed that this behaviour was acquired during carcinoma progression. ECM internalisation, followed by lysosomal degradation, contributed to the up-regulation of the intracellular levels of several amino acids, most notably tyrosine and phenylalanine. This resulted in elevated tyrosine catabolism on ECM under starvation, leading to increased fumarate levels, potentially feeding into the tricarboxylic acid (TCA) cycle. Interestingly, this pathway was required for ECM-dependent cell growth and invasive cell migration under amino acid starvation, as the knock-down of p-hydroxyphenylpyruvate hydroxylase-like protein (HPDL), the third enzyme of the pathway, opposed cell growth and motility on ECM in both 2D and 3D systems, without affecting cell proliferation on plastic. Finally, high HPDL expression correlated with poor prognosis in breast cancer patients. Collectively, our results highlight that the ECM in the tumour microenvironment (TME) represents an alternative source of nutrients to support cancer cell growth by regulating phenylalanine and tyrosine metabolism.

## Introduction

Breast cancer is the most common type of cancer among women. Breast cancer progression starts from benign hyperplasia of epithelial cells of the mammary duct, leading to atypical ductal hyperplasia and ductal carcinoma in situ (DCIS), where cancer cells are surrounded by an intact basement membrane (BM). Eventually, the cells acquire the ability to break through the BM, resulting in the invasive and metastatic cancer phenotype [1].

datasets are deposited in the ORDA repository (https://doi.org/10.15131/shef.data.21608490).

**Funding:** M.N., B.Y. and E.R. are funded by Cancer Research UK (C52879/A29144). M.L.M. is funded by Sheffield/ARAP PhD program. F.B. is funded by 'ARC pour la recherche sur le cancer' Foundation and AMIDEX AMX-2-CE-03 Chaire d'Excellence. The funders had no role in study design, data collection and analysis, decision to publish, or preparation of the manuscript.

**Competing interests:** The authors have declared that no competing interests exist.

**Abbreviations:** BM, basement membrane; CAF, cancer-associated fibroblast; CDM, cell-derived matrix; CME, Costume Module Editor; DCIS, ductal carcinoma in situ; DFBS, dialysed FBS; DMEM, Dulbecco's Modified Eagle's Medium; DMFS, distant metastasis-free survival; ECM, extracellular matrix; EGF, epithelial growth factor; FA, focal adhesion; FAK, focal adhesion kinase; FBS, foetal bovine serum; FDR, false discovery rate; HIF, hypoxia-inducible factor; HPDL, p-hydroxyphenylpyruvate hydroxylase-like protein; MMP, matrix metalloproteinase; MOI, multiplicity of infection; mTORC1, mammalian target of rapamycin complex 1; NF, normal fibroblast; OS, overall survival; PDAC, pancreatic ductal adenocarcinoma; PI, propidium iodide; PPS, palliative performance scale; RFS, relapse-free survival; RT, room temperature; SRSF, Sheffield RNAi Screening Facility; TCA, tricarboxylic acid; TME, tumour microenvironment.

In breast cancer, high breast tissue density is associated with a shift to malignancy and invasion [2,3]. There is growing evidence indicating that the tumour microenvironment (TME) facilitates tumour growth and survival [4]. The TME consists of a variety of cell types, including stromal cells, and extracellular matrix (ECM). The ECM is a highly dynamic 3D network of macromolecules, providing structural and mechanical support to the tissues while interacting with cells through different receptors [5,6].

Due to limited blood supply and elevated tumour growth rate, the TME is often deprived of nutrients, including glucose and amino acids [7]. One of the hallmarks of cancer is the reprogramming of energy metabolism, which provides metabolic flexibility allowing tumour cells to adapt to different nutrient conditions [8]. This includes the ability of cancer cells to benefit from alternative nutrient sources. Indeed, different studies showed that cancer cells benefit from extracellular proteins during food scarcity [9–11]. In Ras-driven pancreatic ductal adenocarcinoma (PDAC) cells, amino acid and glutamine starvation induced albumin and collagen internalisation, respectively, followed by lysosomal degradation and amino acid extraction [7,10,12,13]. Serum and growth factor starvation also stimulated normal mammary epithelial cells to internalise laminin, which resulted in an increase in cellular amino acid content [9]. These studies prompted us to investigate how the internalisation of different components of the ECM impinged on cancer cells' metabolism and growth under amino acid starvation.

To determine whether the ECM surrounding breast cancer cells could provide a metabolically favourable microenvironment, here we assessed the effect of the presence of ECM on breast cancer cell growth and metabolism under glutamine and full amino acid starvation. We demonstrated that the growth of starved invasive breast cancer cells, but not normal mammary epithelial cells, was supported by the presence of different types of ECM. Indeed, Matrigel, collagen I, and CAFs-generated cell-derived matrix (CDM) promoted MDA-MB-231 cell division and enhanced mammalian target of rapamycin complex 1 (mTORC1) activity under amino acid starvation. Interestingly, this was dependent on ECM macropinocytosis mediated by PAK1 and not focal adhesion (FA) signalling. Moreover, metabolomics analyses showed higher amino acid content and up-regulation of phenylalanine and tyrosine metabolism during amino acid deprivation in cancer cells grown on ECM compared to plastic, leading to an increase in fumarate content. Importantly, the down-regulation of an enzyme in this pathway (HPDL) strongly reduced cell growth on ECM under amino acid starvation, without affecting cell growth on plastic. In addition, HPDL and PAK1 knock-down opposed invasive cell migration. Finally, high HPDL expression correlated with poor prognosis in breast cancer patients. Altogether, our data showed that ECM scavenging fuels cancer cell growth by promoting phenylalanine and tyrosine catabolism.

## Results

### The ECM supported invasive breast cancer cell growth under starvation

To assess whether the presence of ECM could promote the survival or growth of invasive breast cancer cells under nutrient deprivation, MDA-MB-231 cells were seeded under complete media, glutamine, or amino acid deficiency and cell growth was quantified on collagen I, Matrigel, and plastic. MDA-MB-231 cells showed significantly higher cell number and growth rate in the presence of collagen I and Matrigel compared to cells on plastic after 8 days of starvation (**Figs 1A–1D** and S1). This represented a partial rescue of cell growth, as the cell numbers under starvation on ECM remained considerably lower than in complete media. Our data also indicated that, in the absence of starvation, cell growth was independent of the ECM, as there was no significant difference in cell number on ECM compared to plastic in complete media (**Figs 1B** and **S1B**). To test the effect of glutamine and amino acid deficiency on cell

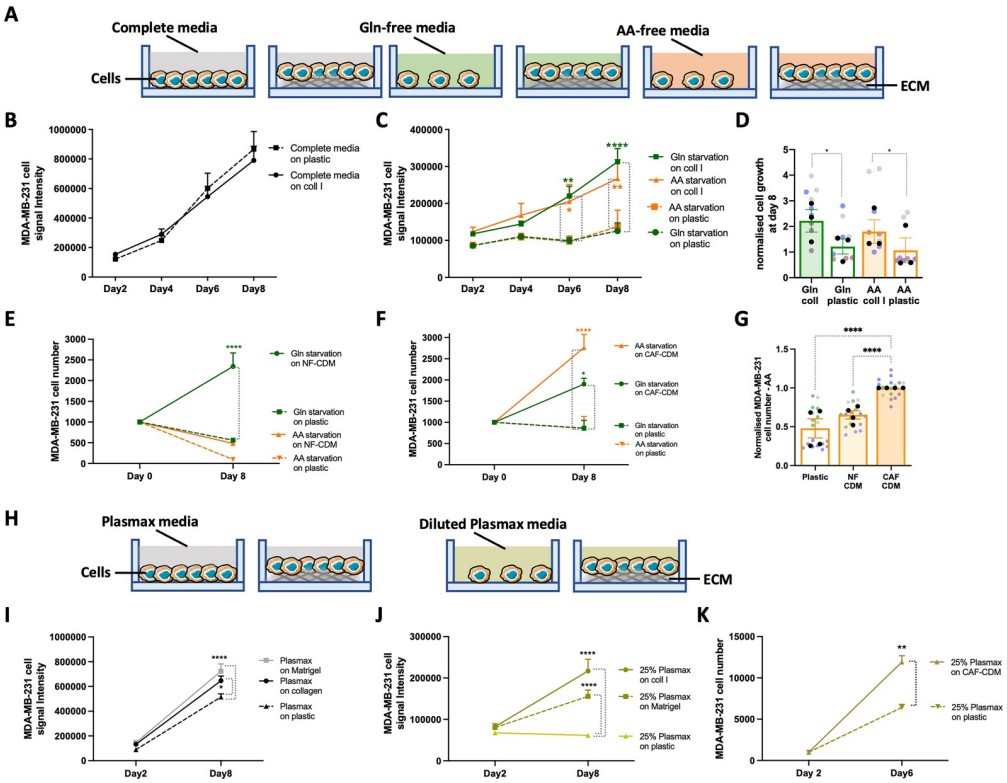

**Fig 1. The ECM supported cell growth under starvation.** (A, H) Schematic, cell proliferation assays. (B–D) MDA-MB-231 cells were seeded on plastic or 2 mg/ml collagen I (coll I) for 8 days under complete media, glutamine (Gln) and amino acid (AA) starvation, fixed, stained with DRAQ5, and imaged with a Licor Odyssey system. Signal intensity was calculated by Image Studio Lite software. (E–G) MDA-MB-231 cells were seeded on plastic, NF-CDM (E, G) or CAF-CDM (F,G) under Gln or AA starvation for 8 days, fixed and stained with Hoechst 33342. Images were collected by ImageXpress micro and analysed by MetaXpress software. (I–K) MDA-MB-231 cells were seeded on plastic, 2 mg/ml collagen I (coll I) or 3 mg/ml Matrigel (I, J) or on CAF-CDM (K) in complete Plasmax media or Plasmax diluted 1:4 in PBS (25%) for 8 or 6 days, fixed and stained with DRAQ5 (I, J) or Hoechst 33342 (K) and quantified as above. Values are mean ± SEM from at least 3 independent experiments (the black dots in the bar graphs represent the mean of individual experiments). *$p < 0.05$, **$p < 0.01$, ****$p < 0.0001$ (C, E, F, I, J, K) two-way ANOVA, Tukey's multiple comparisons test. (D, G) Kruskal–Wallis, Dunn's multiple comparisons test. All the raw data associated with this figure are available in S1 Data. CAF, cancer-associated fibroblast; CDM, cell-derived matrix; ECM, extracellular matrix; NF, normal fibroblast.

growth in a more physiological environment, MDA-MB-231 cells were seeded on CDMs generated by either normal breast fibroblasts or cancer-associated fibroblasts (CAFs) extracted from breast tumours. CDMs are complex 3D matrices that recapitulate several features of native collagen-rich matrices [14]. Different studies highlighted that the CDM produced by CAFs has different structure, composition, and stiffness compared to the CDM derived from normal fibroblasts (NFs) [15–17]. Here, we showed that breast cancer cell growth was partially rescued by both NF- and CAF-generated CDM under glutamine starvation, but only CAF-CDM resulted in a significant increase in cell growth under amino acid deprivation (**Fig 1E–1G**). These data indicate that CAF-CDM provided a more favourable environment for cancer cells' growth compared to NF-CDM under amino acid deprivation and prompted us to study in further detail the mechanisms controlling ECM-dependent cell growth under amino acid, and not glutamine, starvation.

In parallel, to more closely reproduce the in vivo metabolic environment, we used a physiological medium, Plasmax. Plasmax is designed to contain metabolites and nutrients at the

same concentration of human blood to avoid adverse effects of commercial media on cancer cells [18]. Measurement of tumour metabolite content in pancreatic cancer indicated that, on average, the availability was reduced between 50% and 75% [7]. Therefore, to simulate TME starvation conditions, Plasmax was diluted 1:4 with PBS. Cell growth data demonstrated that, while cells on plastic were not able to grow in 25% Plasmax, the presence of collagen I, Matrigel, and CAF-generated CDM resulted in a significant increase in cell numbers (**Fig 1H, 1J and 1K**). Moreover, the ECM had a small but significant effect in promoting cell growth in full Plasmax media as well (**Fig 1I**), suggesting that under more physiological conditions the ECM could support cell growth even in the absence of nutrient starvation. Taken together, these data indicate that the presence of ECM positively affected invasive breast cancer cell growth under nutrient-depleted conditions.

To assess whether ECM-dependent cell growth under starvation was a feature that was acquired during breast cancer progression, we took advantage of the MCF10 series of cells lines [19]. This includes normal mammary epithelial cells (MCF10A, **Fig 2A**), noninvasive

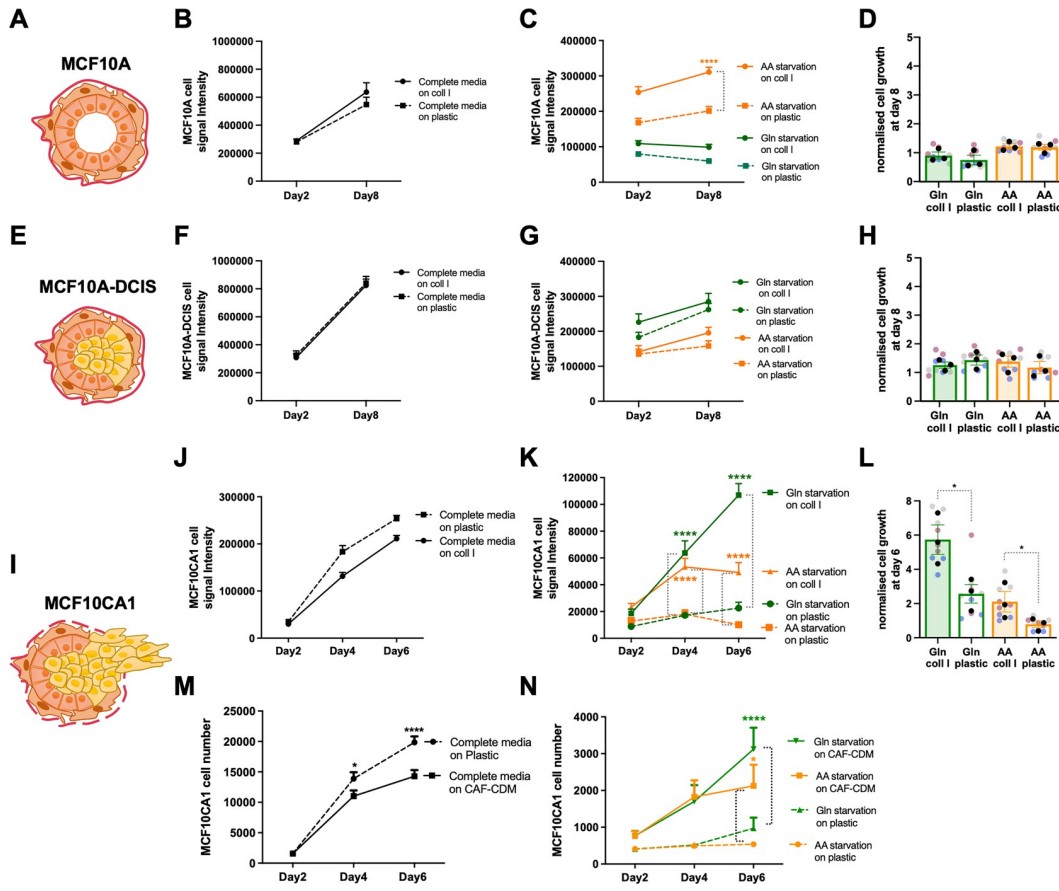

**Fig 2. The ability to use the ECM to promote cell growth was acquired during carcinoma progression.** Schematic, normal mammary gland (A), DCIS (E), and invasive carcinoma (I). MCF10A, MCF10A-DCIS, and MCF10CA1 cells were seeded on plastic, (B–D, F–H, J–L) 2 mg/ml collagen I (coll I) or (M, N) CAF-CDM for 8 or 6 days under complete media, glutamine (Gln) and amino acid (AA) starvation, fixed, stained with DRAQ5 (B–D, F–H, J–L) or Hoechst 33342 (M, N) and imaged with a Licor Odyssey system (B–D, F–H, J–L) or ImageXpress micro (M, N). Signal intensity was calculated by Image Studio Lite software (B–D, F–H, J–L) or MetaXpress software (M, N). Values are mean ± SEM from at least 3 independent experiments (the black dots in the bar graphs represent the mean of individual experiments). *$p < 0.05$, ****$p < 0.0001$ (C, K, M, N) two-way ANOVA, Tukey's multiple comparisons test. (L) Kruskal–Wallis, Dunn's multiple comparisons test. All the raw data associated with this figure are available in S2 Data. CAF, cancer-associated fibroblast; CDM, cell-derived matrix; DCIS, ductal carcinoma in situ; ECM, extracellular matrix.

ductal carcinoma in situ breast cancer cells (MCF10A-DCIS, **Fig 2E**), and metastatic breast cancer cells (MCF10CA1, **Fig 2I**). The cells were seeded either on plastic, collagen I, or Matrigel in glutamine or amino acid-depleted media for 6 or 8 days. In contrast to invasive MDA-MB-231 cells (Fig 1), MCF10A cell growth pattern on collagen I and Matrigel under starvation was similar to the one on plastic. Although cell number was higher on collagen I and Matrigel under amino acid starvation compared to plastic (**Figs 2C and S2C**), there was no significant difference in the growth rate of MCF10A cells on either matrix (**Figs 2D and S2D**). Similarly, our data showed no significant changes in MCF10A-DCIS cell growth on collagen I or Matrigel in amino acid-depleted media compared to the plastic (**Figs 2G and 2H,** S2G and S2H). Interestingly, while collagen I was not able to increase MCF10A-DCIS cell growth under glutamine deficiency (**Fig 2G and 2H**), cell number and growth rate at day 8 post glutamine starvation were significantly higher on Matrigel compared to plastic (**S2G and S2H Fig**). Consistent with our observations in MDA-MB-231 cells, the growth of metastatic MCF10CA1 cells was significantly promoted by collagen I, CAF-CDM, and Matrigel under glutamine or amino acid starvation (**Figs 2J–2N and** S2J–S2L). As before, the presence of ECM did not affect MCF10A, MCF10A-DCIS, and MCF10CA1 cells under complete media, with the exception of Matrigel resulting in a small but significant increase in MCF10A cell growth (**S2B Fig**). These data suggest that the ability of using ECM to compensate for soluble nutrient starvation could have been gradually acquired during cancer progression and could be a defining feature of invasive growth in advanced metastatic breast cancer.

## Collagen I and Matrigel increased cell division under amino acid starvation

The ECM-dependent increase in cell number could be mediated by a stimulation of cell division or an inhibition of cell death, or both. To investigate this, we performed 5-Ethynyl-2'-deoxyuridine (EdU) incorporation experiments, as this thymidine analogue allows the identification of cells which passed through DNA synthesis and mitosis. Cells were starved for 6 days and then incubated with EdU for 2 days (**Fig 3A**). As shown in Fig 3, both glutamine and amino acid starvation significantly reduced the percentage of EdU positive cells on plastic, while the presence of collagen I (**Fig 3B**) or Matrigel (**Fig 3C**) resulted in a significant increase in EdU positive cells. Consistent with the proliferation data, the presence of ECM did not affect EdU incorporation in complete media (**Fig 3B and 3C**). To elucidate whether the ECM also played an anti-apoptotic role, MDA-MB-231 cell death was assessed by measuring the percentage of cells positive for an apoptosis marker, activated caspase-3/7 (**S3A Fig**), after 3 and 8 days of amino acid starvation. Although the apoptosis rate increased between day 3 and day 8 in both complete and amino acid-free media, and amino acid starvation resulted in increased cell death, there was no significant difference between the apoptosis rate on collagen I and Matrigel compared to plastic either at day 3 (**S3B Fig**) or day 8 (**S3C Fig**) in amino acid-depleted media. Similar results were obtained in MCF10CA1 cells after 3 (**S3D Fig**) and 6 (**S3E Fig**) days of amino acid starvation. To rule out the possibility that other cell death mechanisms were affected by the presence of ECM, we performed propidium iodide (PI) staining, a plasma-membrane impermeable dye that has been shown to be incorporated in necrotic cells (**S3F Fig**) [20]. Amino acid starvation significantly increased the percentage of necrotic MDA-MB-231 cells, while the presence of collagen I resulted in a small, and not statistically significant, reduction after both 3 and 8 days of starvation (**S3G and S3H Fig**). In MCF10CA1 cells, PI incorporation was not affected by either nutrient availability or the presence of collagen I (**S3I and S3J Fig**). Thus, our data indicate that collagen I and Matrigel induced invasive breast cancer cell proliferation under amino acid starvation, without having an effect on cell death.

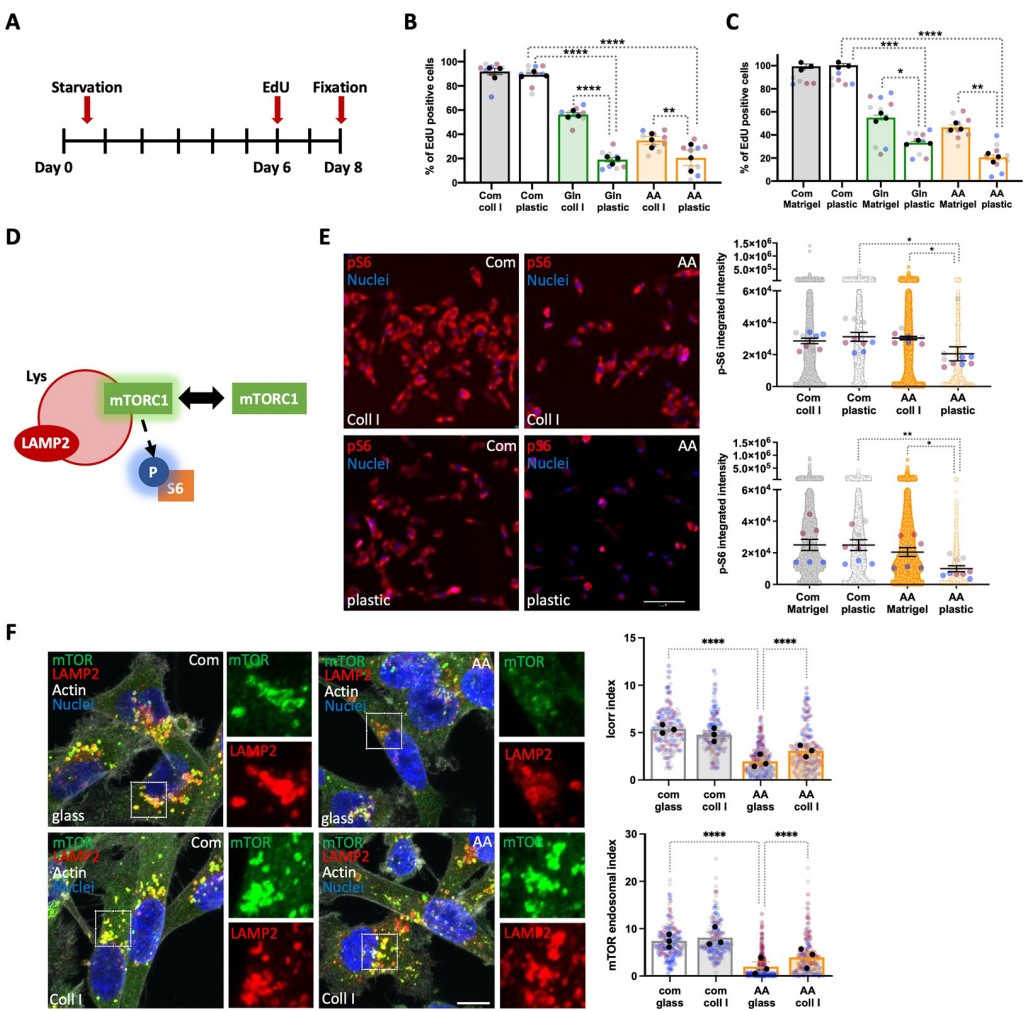

**Fig 3. Collagen I and Matrigel induced cell proliferation and rescued mTORC1 activity under amino acid starvation.** (A) Timeline of EdU incorporation experiments. (B–E) MDA-MB-231 cells were seeded on plastic, (B, E) 2 mg/ml collagen I (coll I) or (C, E) 3 mg/ml Matrigel under complete media (Com), glutamine (Gln) or amino acid (AA) starvation. (B, C) Cells were incubated with EdU at day 6 post starvation, fixed and stained with Hoechst 33342 and Click iT EdU imaging kit at day 8. Images were collected by ImageXpress micro and analysed by MetaXpress software. (D) Schematic, mTORC1 activation on the lysosomal membrane. (E) Cells were fixed and stained for p-S6 and nuclei at day 3 post starvation. Images were collected by ImageXpress micro and analysed by CME software. (F) MDA-MB-231 cells were seeded on uncoated or 1 mg/ml collagen I-coated glass-bottomed dishes for 24 h in complete media, kept in complete (Com) or amino acid-free (AA) media for 24 h, fixed and stained for mTOR (green), LAMP2 (red), actin (white), and nuclei (blue). Samples were imaged with a Nikon A1 confocal microscope. Scale bar, 10 μm. mTOR/LAMP2 co-localisation (Icorr index) and mTOR endosomal accumulation (mTOR endosomal index) were quantified with Image J. The mTOR endosomal index was calculated by thresholding the green channel to identify the objects and plotting the percentage of the ratio between object area and cell area. Values are mean ± SEM from 3 independent experiments (the black dots represent the mean of individual experiments). $^{*}p < 0.05$, $^{**}p < 0.01$, $^{***}p < 0.001$, $^{****}p < 0.0001$ Kruskal–Wallis, Dunn's multiple comparisons test. All the raw data associated with this figure are available in S3 Data. mTORC1, mammalian target of rapamycin complex 1.

## Collagen I and Matrigel rescued mTORC1 activity in starved cells

The mammalian target of Rapamycin (mTOR) signalling pathway is a key regulator of anabolic and catabolic processes. The presence of amino acids, glucose, and growth factors activates mTOR complex 1 (mTORC1), which triggers downstream anabolic signalling pathways. However, the lack of amino acids and growth factors deactivates mTORC1 and induces

catabolism, resulting in lysosome biogenesis and autophagy [21]. Therefore, we investigated whether higher proliferation rate of cells on collagen I and Matrigel was accompanied by a higher mTORC1 activity in starved MDA-MB-231 cells. We examined the phosphorylation of the ribosomal subunit S6, a key signalling event downstream of mTORC1 activation (**Fig 3D**).

MDA-MB-231 cells were starved for 3 days either on collagen I, Matrigel, or plastic, fixed and stained for phospho-S6 (p-S6). Quantifying the intensity of p-S6, as expected, demonstrated that amino acid starvation resulted in a significant reduction in mTORC1 activity on plastic. Interestingly, the presence of ECM fully rescued mTORC1 activation under amino acid starvation (**Fig 3E**). Since we found that collagen I failed to rescue cell proliferation in MCF10A and MCF10A-DCIS cells, we investigated whether the re-activation of mTOR was sufficient to promote their growth on collagen I under amino acid starvation. Despite treatment of MCF10A cells with the mTOR activator MHY1485 [22] resulted in a small, but statistically significant, increase in the levels of phosphorylation of the mTOR downstream regulator 4EBP1 (**S4A Fig**), we did not observe any statistically significant change in the growth of MCF10A and MCF10A-DCIS cells on collagen I under amino acid starvation (**S4B Fig**), suggesting that the activation of mTORC1 is not sufficient to promote cell proliferation under starvation. mTOR recruitment to the lysosomal membrane has been shown to be integral to mTORC1 activation, as the mTORC1 complex senses amino acid availability in the lysosomes [23]. In agreement with our p-S6 data, amino acid starvation resulted in a significant reduction of mTOR lysosomal recruitment, quantified as co-localisation with the lysosomal marker LAMP2, and mTOR endosomal accumulation, quantified as the area covered by mTOR positive endosomes. Interestingly, the presence of collagen I led to a significant increase in mTOR lysosomal targeting under amino acid starvation, without affecting mTOR distribution in complete media (**Fig 3F**). Together, given the fact that amino acid availability in lysosomes can promote mTOR recruitment and activation, we can speculate that ECM endocytosis and degradation might result in higher lysosomal amino acid concentrations, supporting mTORC1 translocation and activation.

## Inhibition of ECM internalisation opposed cell growth under amino acid starvation

It was previously shown that, under starvation, cancer cells relied on scavenging extracellular proteins to maintain their survival and growth [9,13]. First, we characterised the ability of MCF10A, MCF10A-DCIS, and MCF10CA1 cells to internalise ECM and found that collagen I and CDM internalisation was strongly induced in metastatic breast cancer cells, compared to noninvasive breast cancer cells and non-transformed mammary epithelial cells (**Fig 4A and 4B**). To examine whether invasive breast cancer cells had the ability to internalise ECM components under starvation, we tracked the ECM journey inside the cells in the presence and absence of a lysosomal protease inhibitor (E64d), to prevent lysosomal degradation. E64d has been previously shown to promote the intracellular accumulation of lysosomally delivered proteins, including internalised collagen IV [24]. The uptake of fluorescently labelled collagen I (**S5A Fig**) and Matrigel (**S5B Fig**) was monitored in complete or amino acid-depleted media. The quantification of collagen I and Matrigel uptake demonstrated that, under all nutrient condition, MDA-MB-231 cells significantly accumulated higher amounts of collagen I and Matrigel inside the cells when lysosomal degradation was inhibited (**S5A and S5B Fig**), indicating that ECM components undergo lysosomal degradation following internalisation.

We then wanted to investigate whether ECM-dependent cancer cell growth under nutrient deficiency relied on ECM internalisation. To assess this, collagen I and Matrigel-coated plates were treated with 10% glutaraldehyde to chemically crosslink the amine groups of ECM

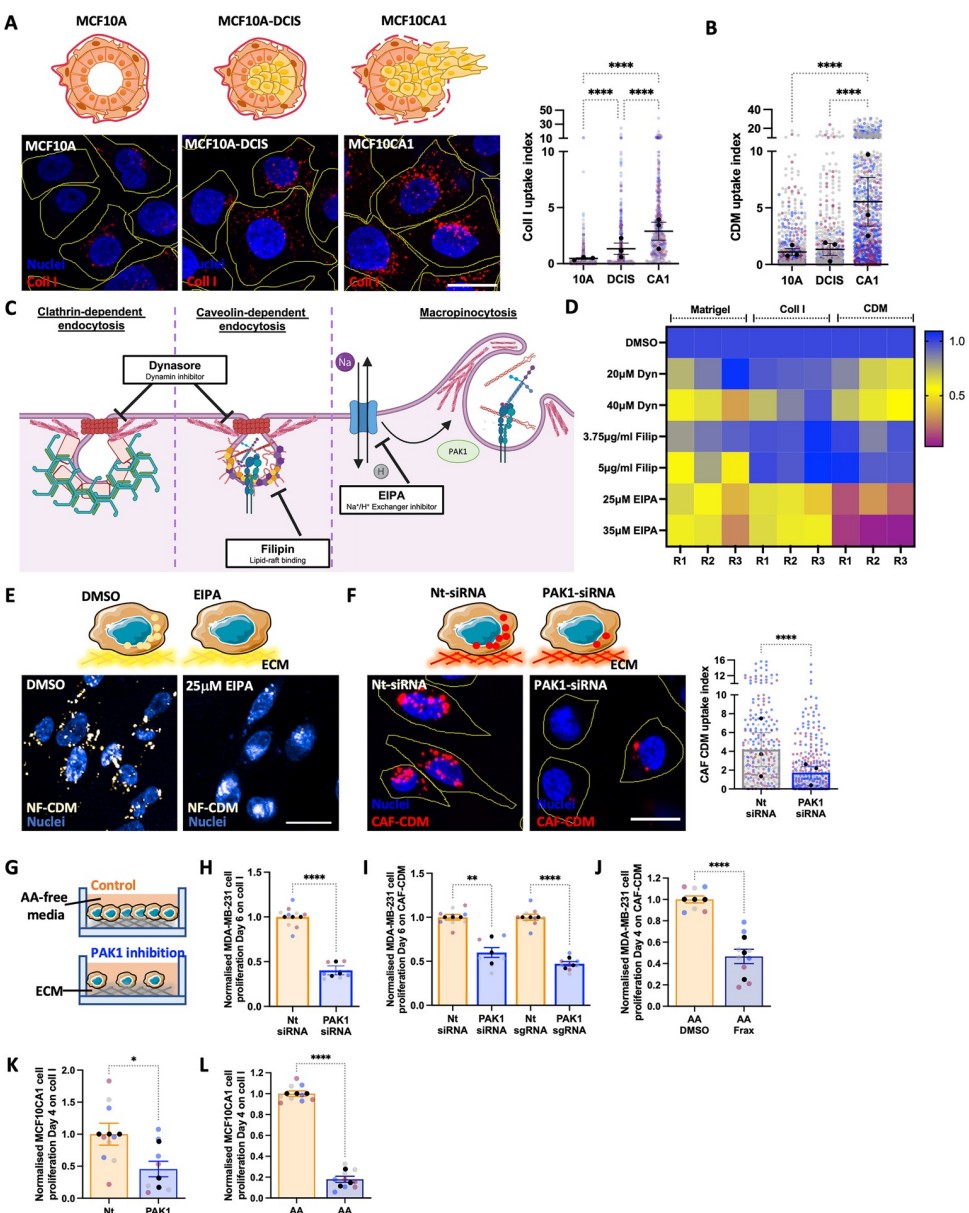

**Fig 4. PAK1-mediated ECM macropinocytosis was required for ECM-dependent cell growth under amino acid starvation.** MCF10A (10A), MCF10A-DCIS (DCIS), and MCF10CA1 (CA1) were seeded on pH-rodo-labelled 1 mg/ml collagen I (A) or NF-CDM (B) for 6 h, stained with Hoechst 33342 (blue) and imaged live with a Nikon A1 confocal microscope. Scale bar, 20 μm. ECM uptake index was quantified with Image J. (C) Schematic, main endocytic pathways and inhibitor targets. (D, E) MDA-MB-231 cells were seeded on pH-rodo labelled 0.5 mg/ml Matrigel, 0.5 mg/ml collagen I (coll I), or NF-CDM (CDM, white) in the presence of DMSO control, 20 μm or 40 μm dynasore, 3.75 μg/ml or 5 μg/ml filipin, 25 μm or 35 μm EIPA for 6 h, stained with Hoechst 33342 (blue), imaged live with an Opera Phenix microscope and analysed with Columbus software. Scale bar, 20 μm. (F) MDA-MB-231 cells were transfected with a siRNA targeting PAK1 (PAK1-siRNA) or a non-targeting siRNA control (nt-siRNA), plated on pH-rodo labelled CAF-CDM (red) for 6 h, stained with Hoechst 33342 (blue) and imaged live with a Nikon A1 confocal microscope. Scale bar, 20 μm. ECM uptake index was quantified with Image J. (G) Schematic, cell proliferation experiments. MDA-MB-231 (H, I) and MDA-MB-231 CRSPRi (I) cells were plated on 2 mg/ml collagen I (coll I, H) or CAF-CDM (I), transfected with a siRNA targeting PAK1 (PAK1-siRNA), a non-targeting siRNA control (nt-siRNA, H,I), a synthetic guide RNA targeting PAK1 (PAK1-sgRNA) or a non-targeting synthetic guide RNA control (nt-sgRNA, I) and cultured under amino acid starvation for 6 days. MDA-MB-231 cells (J) and MCF10CA1 cells (L) were grown on CAF-CDM (J) or 2 mg/ml collagen I (coll I, L) under amino acid (AA) starvation for 4 days in the presence of 3 μm (J) or 1 μm (L) FRAX597, fixed and stained with Hoechst 33342. (K) MCF10CA1 cells were transfected with a siRNA targeting PAK1 (PAK1-siRNA) or a non-targeting siRNA control (nt-siRNA) and cultured under amino acid

(AA) starvation for 4 days. Cells were fixed and stained with Hoechst 33342. Images were collected by ImageXpress micro and analysed by MetaXpress software. Values are mean ± SEM and from 3 independent experiments (the black dots represent the mean of individual experiments). *$p < 0.05$, **$p < 0.01$, ****$p < 0.0001$ Mann–Whitney test (F, H, J–L) or Kruskal–Wallis, Dunn's multiple comparisons test (A, B, I). All the raw data associated with this figure are available in S4 Data. CAF, cancer-associated fibroblast; CDM, cell-derived matrix; DCIS, ductal carcinoma in situ; ECM, extracellular matrix; NF, normal fibroblast.

components. Our data confirmed that cross-linking completely opposed collagen I and Matrigel internalisation in both MDA-MB-231 and MCF10CA1 cells (S5C–S5E Fig). Interestingly, while the growth of MDA-MB-231 cells was not affected by matrix cross-linking in complete media (S5F Fig), the lack of collagen I and Matrigel uptake due to cross-linking completely opposed cell growth under amino acid starvation in both MDA-MB-231 and MCF10CA1 cells (S5G and S5H Fig). High matrix cross-linking has been shown to also oppose extracellular ECM degradation mediated by matrix metalloproteinases (MMPs) [25]; therefore, we wanted to determine whether the inhibition of ECM-dependent cell growth by cross-linked matrices was due to a reduction in ECM degradation by MMPs (S5I Fig). In agreement with previous reports showing that ECM internalisation was not dependent on MMP activity [26], our data indicated that MMP inhibition by treatment with the broad-spectrum inhibitor GM6001 did not reduce collagen I and Matrigel uptake (S5J Fig). Consistent with this, MMP inhibition did not affect cell growth under complete media or amino acid starvation on both plastic and collagen I (S5K–S5M Fig), suggesting that the lack of cell growth observed on cross-linked matrices is due to their inability to be internalised by breast cancer cells.

We therefore set out to characterise the mechanisms controlling ECM internalisation in breast cancer cells. Several endocytic pathways, including clathrin-dependent, caveolin-dependent endocytosis and macropinocytosis have been previously implicated in the internalisation of different ECM components [26]. We developed a high-throughput imaging method, based on pH-rodo labelled ECM and live cell imaging. pH-rodo is a pH-sensitive dye, whose fluorescence is minimal at neutral pH but strongly increases in the acidic environment of late endosomes and lysosomes. It has been extensively used in phagocytosis studies [27,28]. As a first step, we used pharmacological inhibitors targeting dynamin, a GTPase involved in both clathrin- and caveolin-mediated endocytosis (dynasore), lipid-raft-mediated endocytosis, which includes caveolin-dependent internalisation (filipin) and macropinocytosis (EIPA, Fig 4C) and we assessed the internalisation of Matrigel, collagen I, and NF-generated CDM by MDA-MB-231 cells. As shown in Fig 4D and 4E, the strongest uptake inhibition of all ECM tested was obtained by EIPA treatment, while dynasore decreased Matrigel and CDM endocytosis, but not collagen I and filipin only opposed Matrigel uptake. To validate these data, we used siRNA-mediated down-regulation of key endocytosis regulators and assessed Matrigel uptake. Consistent with the inhibitors' results, the knock-down of dynamin 2/3, caveolin 1/2 and PAK1, a key macropinocytosis regulator [29], all resulted in a reduction of Matrigel endocytosis (S6A Fig). Similarly, caveolin 1/2 and dynamin 2/3 down-regulation significantly impaired CDM uptake (S6B Fig), indicating that multiple endocytic pathways, including macropinocytosis, are likely responsible for the internalisation of complex ECM. Indeed, PAK1 knockdown (S6C Fig) strongly impaired CAF-CDM (Fig 4F), NF-CDM, and collagen I uptake in MDA-MB-231 cells (S6D Fig). To investigate whether the increased ECM uptake we observed in MCF10CA1 cells was due to a higher PAK1 expression in these cells, compared to MCF10A and MCF10A-DCIS cells, we measured PAK1 protein levels, but did not detect any correlation between PAK1 expression and ECM internalisation ability (S6E Fig).

Since our data indicate that macropinocytosis is required for ECM endocytosis in breast cancer cells, we wanted to investigate whether PAK1-mediated ECM uptake was required for

ECM-dependent cell growth. Consistent with the matrix cross-linking data, PAK1 down-regulation, obtained by either siRNA (**Fig 4G–4I**) or CRISPRi sgRNA (**Fig 4I**) strongly reduced cell growth under amino acid starvation when cells were seeded on collagen I or CAF-CDM. In addition, treatment with an ATP-competitive PAK inhibitor (FRAX597, **Fig 4J**) also prevented ECM-dependent cell growth, indicating that PAK1 catalytic activity is required for this. Interestingly, PAK1 inhibition did not affect cell proliferation on plastic, while resulted in a small but significant reduction in cell growth on ECM in the presence of complete media (**S6F–S6H Fig**). Similarly, PAK pharmacological inhibition and PAK1 down-regulation substantially reduced MCF10CA1 cell growth on collagen I under amino acid starvation (**Fig 4K and 4L**). Altogether, these results demonstrate that macropinocytosis-dependent ECM internalisation is necessary for ECM-dependent breast cancer cell growth under amino acid starvation.

## Inhibition of adhesion signalling did not affect ECM-dependent cell growth under starvation

ECM cross-linking has been shown to increase matrix stiffness, thereby affecting cell adhesion and integrin signalling [26]. Binding of ECM components to the integrin family of ECM receptors induces the recruitment of focal adhesion (FA) proteins on the integrin intracellular domain, triggering downstream signalling pathways (**Fig 5A**) [30,31]. FA kinase (FAK) is a non-receptor tyrosine kinase located in FAs involved in the regulation of several cell behaviours, including proliferation, migration, and survival. FAK is also overexpressed in some cancers, and it has been linked to their aggressiveness [32]. Cell-ECM interaction triggers FAK auto-phosphorylation on Tyr397, which exposes an SH2 domain-binding site for the protein kinase Src. Further Src-dependent FAK phosphorylation results in full FAK activation [33]. To test whether the inhibition of FA signalling affected cell growth under nutrient starvation, MDA-MB-231 cells were treated with a FAK inhibitor, PF573228. Firstly, the auto-phosphorylation of FAK Tyr397 was measured in the presence of different PF573228 concentrations, to assess the effectiveness of the inhibitor. Our results revealed that around 80% inhibition of FAK activity was achieved with 0.5 μm PF573228 (**Fig 5B and 5C**). Therefore, to assess the effect of FAK inhibition on cell growth, MDA-MB-231 cells were seeded either on collagen I, Matrigel, or CAF-generated CDM under complete media or amino acid starvation in the presence of 0.5 μm PF573228. Interestingly, we did not observe any significant difference in cell number between the control group and cells treated with the FAK inhibitor, either in complete media or amino acid deprivation (**Fig 5D–5F**). Similarly, the treatment of MDA-MB-231 cells with the Src inhibitor PP2 did not affect cell growth on either collagen I or CAF-generated CDM under amino acid starvation, while significantly reduced cell growth in complete media, in agreement with previous observations in the literature [34] (**Fig 5G–5J**). In addition, staining for the FA protein paxillin indicated no changes in the quantity and distribution of FAs in cells under amino acid starvation compared to the complete media, on either collagen I or Matrigel (**Fig 5K and 5L**), or on cross-linked matrices (**Fig 5M**). Taken together, these data indicate that FAK- and Src-dependent signalling events following cell-ECM interaction did not affect cell growth under amino acid starvation and ECM-dependent cell growth was predominantly mediated by ECM internalisation followed by lysosomal degradation.

Increased ECM stiffness has been associated with cancer progression, leading to altered cell proliferation and metabolism [35]. Therefore, we used different rigidities to test whether ECM uptake and ECM-dependent cell proliferation were stiffness-dependent. Non-enzymatic glycation by reducing sugars, such as ribose, has been shown to result in increased collagen I stiffness (from 0.2 kPa to 0.8 kPa in the presence of 200 mM ribose) [36]. Therefore, we tested

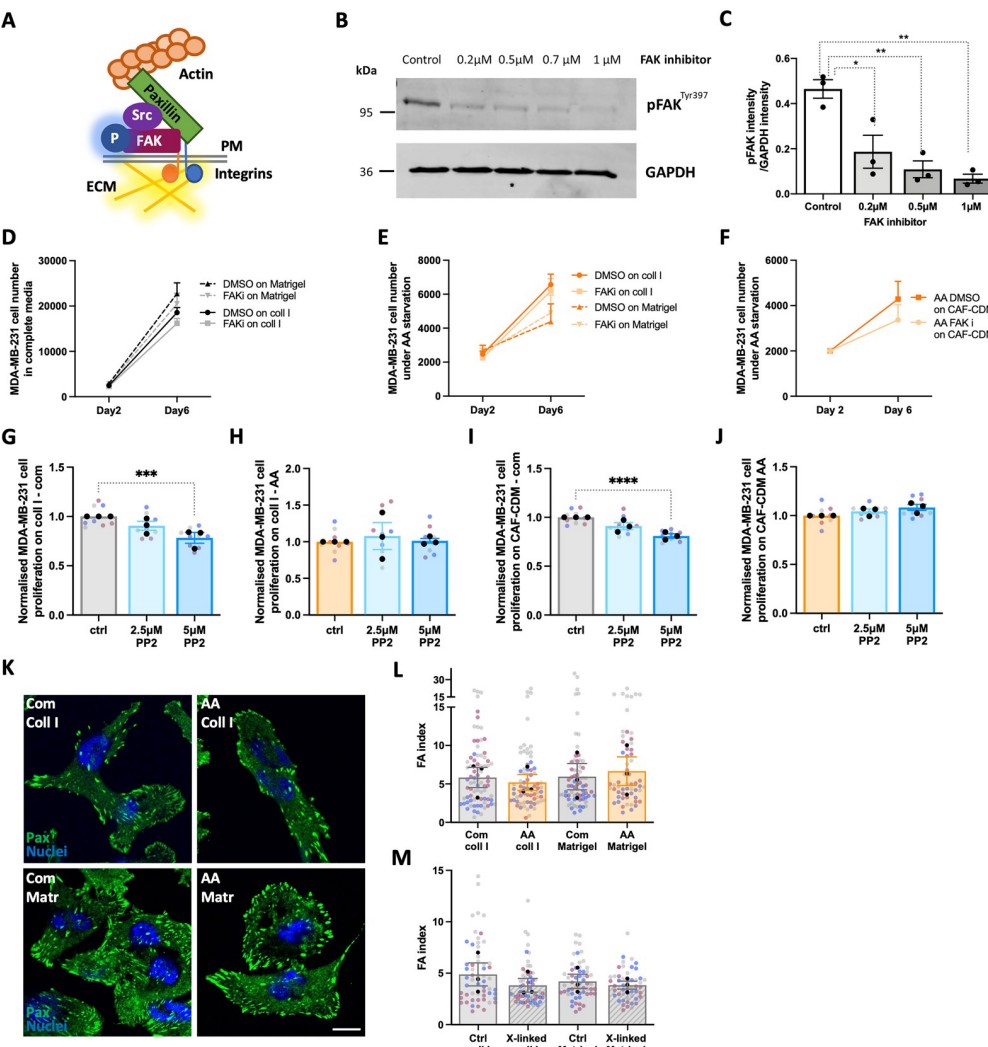

**Fig 5. FA signalling was not required for ECM-dependent cell growth.** (A) Schematic, FAs. (B, C) MDA-MB-231 cells were suspended in complete media containing either 0.2 μm, 0.5 μm, 0.7 μm, or 1 μm PF573228 (FAK inhibitor) or DMSO (vehicle) for 30 min. Cells were seeded on 0.5 mg/ml collagen I for 30 min in the same media, lysed and proteins analysed by western blotting. The intensity of the bands was quantified using the Licor Odyssey system. Values are mean ± SEM from 3 independent experiments. *$p < 0.05$, **$p < 0.01$ Kruskal–Wallis, Dunn's multiple comparisons test. (D–F) MDA-MB-231 cells were seeded on either 2 mg/ml collagen I (coll I), 3 mg/ml Matrigel, CAF-CDM or plastic under complete media or amino acid starvation for 6 days and treated with 0.5 μm PF573228 (FAKi) or DMSO (control) every 2 days. Cells were fixed and stained with Hoechst 33342. Images were collected by ImageXpress micro and analysed by MetaXpress software. Values are mean ± SEM from at least 3 independent experiments. (G–J) MDA-MB-231 cells were seeded on 2 mg/ml collagen I (coll I), CAF-CDM, or plastic under complete media or amino acid starvation for 4 days and treated with 2.5 μm, 5 μm PP2 or DMSO (control) every 2 days, stained and analysed as in (D). Values are mean ± SEM from 3 independent experiments. ***$p < 0.001$, ****$p < 0.0001$ Kruskal–Wallis, Dunn's multiple comparisons test. (K–M) MDA-MB-231 cells were plated on 2 mg/ml collagen I (coll I) or 3 mg/ml Matrigel (Matr), previously cross-linked with 10% glutaraldehyde where indicated (x-linked), for 3 days in complete (Com) or amino acid-free (AA) media, fixed and stained for paxillin (green) and nuclei (blue). Bar, 10 μm. Images were acquired with a Nikon A1 confocal microscope and quantified with Image J. Values are mean ± SEM from 3 independent experiments (the black dots represent the mean of individual experiments). All the raw data associated with this figure are available in S5 Data. CAF, cancer-associated fibroblast; CDM, cell-derived matrix; ECM, extracellular matrix; FAK, focal adhesion kinase.

whether ribose cross-linking affected collagen I internalisation and cell growth, but did not detect any statistically significant difference in collagen uptake or collagen I-dependent cell proliferation when comparing ribose cross-linked with control collagen I (**S7A and S7B Fig**). Similarly, we measured the proliferation of cells plated on poly-acrylamide hydrogels of different stiffnesses by performing EdU incorporation and found that ECM-dependent cell growth under amino acid starvation was slightly reduced as stiffness increased (**S7C Fig**). Together, these data suggest that mechanotransduction is not a key regulator of ECM-dependent invasive breast cancer cell growth under amino acid starvation.

## ECM-dependent cell growth was mediated by phenylalanine/tyrosine metabolism under amino acid starvation

It has been previously demonstrated that laminin endocytosis resulted in an increase in intracellular amino acid levels in serum starved epithelial cells [9]; therefore, we hypothesised that metabolites derived from ECM internalisation and degradation would result in changes in cellular metabolome. To quantify ECM-dependent differences in the cellular metabolite content, we performed non-targeted direct infusion mass spectrometry of MDA-MB-231 cells either on plastic or CAF-CDM under amino acid starvation for 6 days (**Fig 6A**). On CAF-CDM, we detected 716 up-regulated and 749 down-regulated metabolites (**Fig 6B**). Metabolomic pathway analysis revealed that phenylalanine and tyrosine metabolism and the pentose phosphate pathway were the most significantly enriched (**Fig 6C**). We then used a similar approach comparing MDA-MB-231 cells grown on plastic, collagen I, or Matrigel for 6 days under amino acid starvation (**S8A, S8B and S8D Fig**). Metabolic pathway analysis revealed that purine metabolism and phenylalanine/tyrosine metabolism were the most up-regulated (**S8B–S8E Fig**). Consistently, when we assessed amino acid intracellular levels, we found that phenylalanine and tyrosine were the most up-regulated on all the ECMs tested compared to plastic under amino acid starvation (**Fig 6D**). By performing a metabolic pathway analysis restricted to the metabolites commonly up-regulated on all ECMs compared to plastic (**Fig 6E**), tyrosine and phenylalanine metabolism was identified as the most up-regulated pathway (**Fig 6F**).

Phenylalanine metabolism is composed of a series of metabolic reactions, leading to the production of fumarate, which can enter the tricarboxylic acid (TCA) cycle, leading to energy production (**Fig 6G**). Using targeted ultra-performance liquid chromatography-tandem mass spectrometry in MDA-MB-231 cells grown under amino acid starvation for 6 days, we confirmed that the presence of CAF-CDM and collagen I resulted in increased levels of phenylalanine, tyrosine, and fumarate (**Fig 6H and 6I**). Interestingly, the inhibition of ECM uptake by treatment with the PAK inhibitor FRAX597 resulted in a reduction of phenylalanine and tyrosine in cells grown on collagen I under amino acid starvation (**Fig 6J**), suggesting that ECM endocytosis is required for the up-regulation of tyrosine and phenylalanine on ECM under starvation. We reasoned that, if the ECM supported cell growth by promoting phenylalanine catabolism, the knock-down of a central enzyme in this pathway would inhibit ECM-dependent cell growth under amino acid deprivation (**Fig 6K**). To test this, we transiently knocked-down p-hydroxyphenylpyruvate hydroxylase (HPD) and its paralogue p-hydroxyphenylpyruvate hydroxylase-like protein (HPDL) and we assessed cell proliferation on plastic or CAF-CDM, under complete media or amino acid starvation. In both cases, we observed a significant reduction in protein or mRNA expression upon siRNA transfection (**S8O and S8P Fig**). Strikingly, HPDL knockdown, but not HPD, significantly impaired cell growth under amino acid starvation on CAF-CDM (**Fig 6L**) and on collagen I (**S8G Fig**).

Interestingly, both HPD and HPDL knock-down resulted in a small, albeit statistically significant, reduction in cell proliferation on CDM in complete media and opposed cell growth

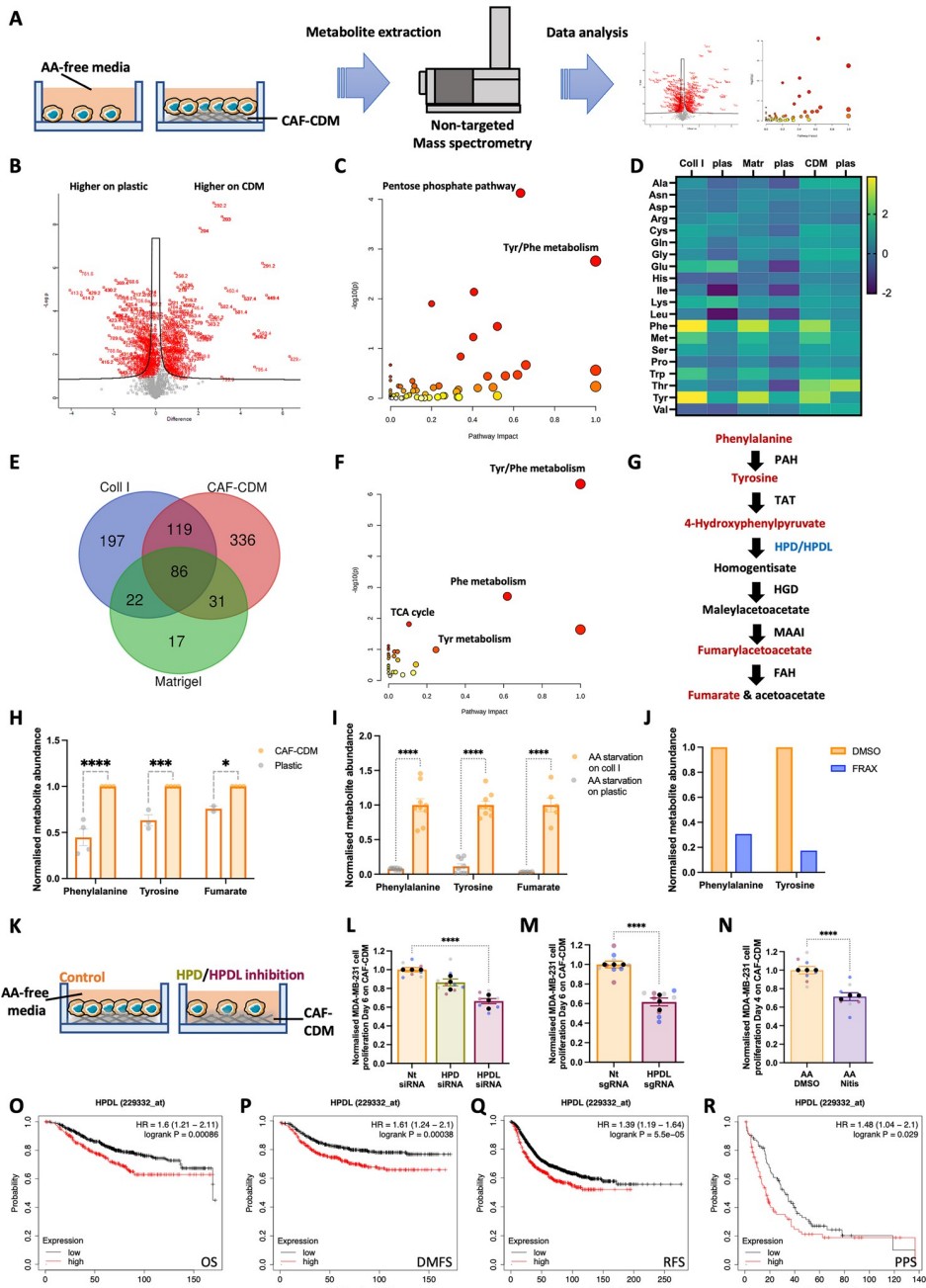

**Fig 6. ECM-dependent cell growth was mediated by tyrosine catabolism.** (A) Metabolomics workflow. MDA-MB-231 cells were plated on plastic or CAF-CDM for 6 days in amino acid-free media. Metabolites were extracted and quantified by non-targeted mass spectrometry. Volcano plot (B), enriched metabolic pathways (C) and changes in amino acid levels (D) are presented. This dataset is deposited in ORDA (https://doi.org/10.15131/shef.data.21608490). (E) Venny diagram and metabolic pathway analysis (F) of the statistically significant up-regulated metabolites in cells plated on 2 mg/ml collagen I (coll I), CAF-CDM, or 3 mg/ml Matrigel under amino acid starvation for 6 days. (G) Schematic, phenylalanine catabolism. (H–J) Metabolites were prepared as in (A), or in the presence of 2 μm FRAX597, and the levels of phenylalanine, tyrosine, and fumarate were measured by targeted mass spectrometry. (K, L) MDA-MB-231 cells were plated on CAF-CDM, transfected with siRNA targeting HPD (HPD siRNA), siRNA targeting HPDL (HPDL siRNA), or non-targeting siRNA control (Nt siRNA) and cultured under amino acid (AA) starvation for 6 days. Cells were fixed and stained with Hoechst 33342. Images were collected by ImageXpress micro and analysed by MetaXpress software. (M) MDA-MB-231 CRSPRi cells were plated on CAF-CDM, transfected with a synthetic guide RNA targeting HPDL (HPDL sgRNA) and a non-targeting synthetic guide RNA control (Nt sgRNA) and analysed as in (L). (N) MDA-MB-231 cells were grown on CAF-CDM under amino acid (AA) starvation for 4 days in the presence of 40 μm Nitisinone (Nitis) and analysed as in (L). *$p < 0.05$, ***$p < 0.001$, ****$p < 0.0001$ two-way

ANOVA, Tukey's multiple comparisons test (H, I), Mann–Whitney test (N, M) or Kruskal–Wallis, Dunn's multiple comparisons test (L). (O–R) 943 breast cancer patients were stratified into low and high HPDL expressors based on median gene expression. The Kaplan–Meier analysis compared OS, DMFS, RFS, and PPS of patients with tumours expressing high levels of HPDL (red), with those expressing low HPDL levels (black). The numerical data associated with this figure are available in S6 Data. CAF, cancer-associated fibroblast; CDM, cell-derived matrix; DMFS, distant metastasis-free survival; ECM, extracellular matrix; HPDL, p-hydroxyphenylpyruvate hydroxylase-like protein; OS, overall survival; RFS, relapse-free survival.

in diluted Plasmax media (**S8F and S8H Fig**). In addition, HPD down-regulation, but not HPDL, modestly reduced cell numbers on plastic (**S8I Fig**). Consistently, CRISPRi-mediated HPDL down-regulation opposed cell growth under amino acid starvation on CAF-CDM (**Fig 6M**). Both HPD and HPDL have previously been shown to be involved in multiple metabolic pathways [37,38]. To assess whether HPDL was controlling phenylalanine catabolism in our settings, we measured intracellular fumarate levels and we showed that HPDL knock-down resulted in a significant reduction in fumarate concentration (**S8J Fig**). To confirm that fumarate could be generated by tyrosine catabolism, we performed an isotope labelling experiment by incubating MDA-MB-231 cells with C13-tyrosine. Interestingly, we detected a higher proportion of C13-labelled fumarate on plastic than on collagen I (**S8K Fig**), suggesting that tyrosine derived from collagen I degradation could contribute to fumarate production, therefore, increasing the M+0 pool in cells plated on collagen I. To confirm that HPD/HPDL catalytic activity was required for ECM-dependent cell growth, we treated MDA-MB-231 cells with nitisinone, an FDA-approved 4-hydroxyphenylpyruvate dioxygenase inhibitor currently used to treat hereditary tyrosinemia type 1 and observed a significant reduction in cell numbers on CAF-CDM under amino acid starvation (**Fig 6N**). Consistent with these data, metabolomics analysis identified tyrosine and phenylalanine metabolism among the up-regulated pathways in MCF10CA1 cells grown on collagen I under amino acid starvation (**S8L and S8M Fig**). Targeted ultra-performance liquid chromatography-tandem mass spectrometry confirmed increased phenylalanine, tyrosine, and fumarate levels in MCF10CA1 cells on collagen I under amino acid starvation, compared to plastic (**S8N Fig**) and HPDL down-regulation resulted in a significant inhibition of cell proliferation (**S8Q Fig**). Given the requirement for HPDL to control ECM-dependent cell growth under amino acid starvation, we looked at the relationship between HPDL expression and disease outcome in breast cancer. High expression of HPDL was associated with significantly decreased overall survival (OS), distant metastasis-free survival (DMFS), relapse-free survival (RFS), and palliative performance scale (PPS) (**Fig 6O–6R**). Altogether, these data indicate that HPDL has a key role in controlling ECM-dependent cell growth under amino acid starvation.

3D spheroids have been shown to display significant metabolic changes compared to their 2D counterparts [39]; therefore, we assessed whether the ECM was also able to sustain cell proliferation in 3D culture conditions. We first measured the ability of MDA-MB-231 cells grown as spheroids to internalise ECM components, by labelling the collagen I and geltrex mixture with pH-rodo. Live cell imaging demonstrated a significant increase in ECM endocytosis 48 h and 72 h after amino acid starvation, suggesting that nutrient deprivation promotes ECM uptake in 3D systems (**Fig 7A and 7B**). Moreover, we investigated whether tyrosine catabolism was required for the growth of MDA-MB-231 cell spheroids under amino acid starvation. Consistent with our 2D experiments, we observed that amino acid starved spheroids retained substantial EdU incorporation, while the presence of the HPD/HPDL inhibitor nitisinone resulted in a statistically significant reduction in the percentage of EdU positive cells (**Fig 7C**). Together with our 2D results, these data demonstrate that tyrosine catabolism plays a central role in promoting ECM-dependent cancer cell proliferation, in both 2D and 3D environments.

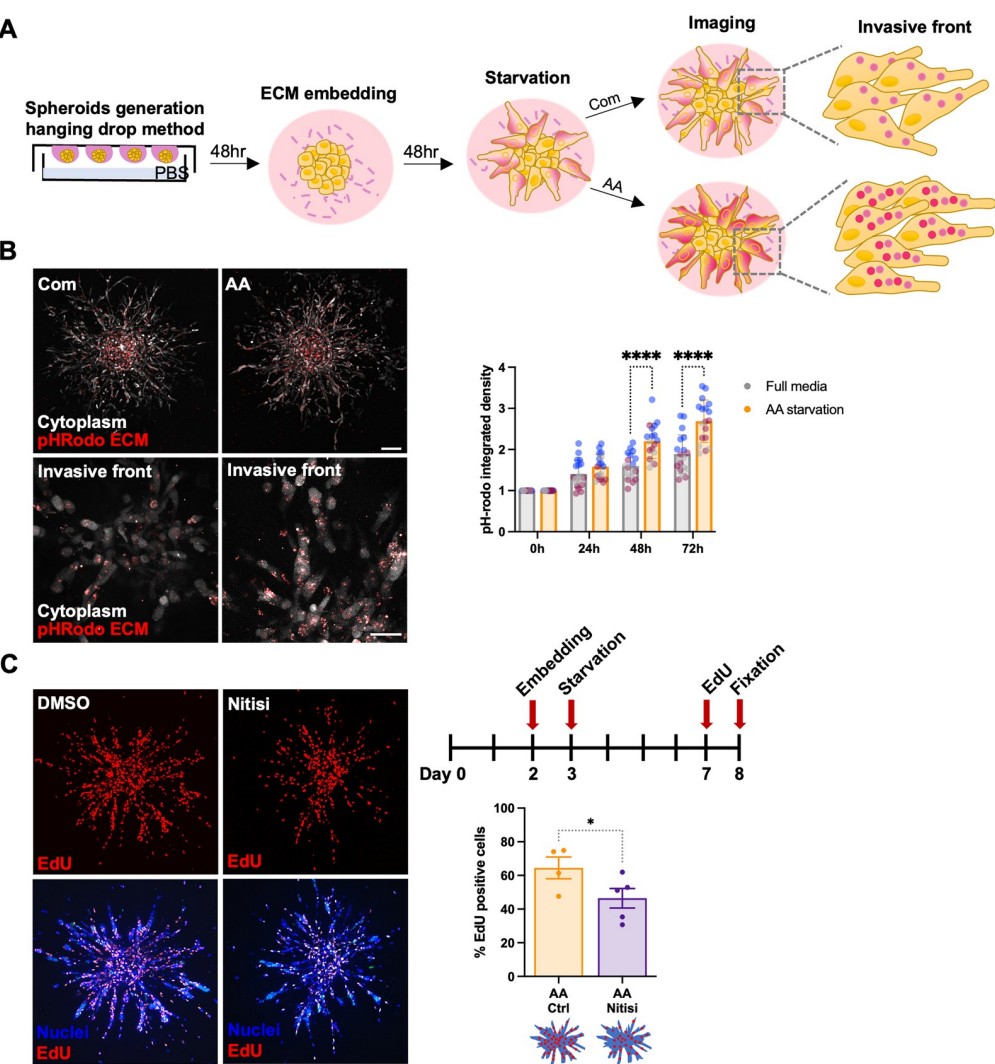

**Fig 7. Tyrosine catabolism supported cell growth in 3D systems.** (A) Schematic, ECM uptake in 3D spheroids. (B) MDA-MB-231 cell spheroids were generated by the hanging drop method, embedded in 3 mg/ml pH-rodo labelled collagen I and geltrex (50:50) mixture for 2 days, starved of amino acid (AA) or kept in complete media (Com) for 2 days and imaged live with a Nikon A1 confocal microscope. Representative images of whole spheroids and higher magnification of the invasive front are shown. Scale bar, 132 μm and 42 μm, respectively. Values are mean ± SD from 3 independent experiments, ****$p < 0.0001$ two-way ANOVA, Tukey's multiple comparisons test. (C) MDA.MB-231 cell spheroids were generated as in (B) in unlabelled 3 mg/ml collagen I and Geltrex (50:50) mixture, starved in amino acid-free (AA) media for 5 days, incubated with EdU for 1 day, fixed and stained with Hoechst 33342 and Click iT EdU imaging kit at day 6. The percentage of EdU-positive cells was measured with Image J. Values are mean ± SEM from 3 independent experiments *$p = 0.0286$ Mann–Whitney test. All the raw data associated with this figure are available in S7 Data. ECM, extracellular matrix.

## PAK1 and HPDL were required for cancer cell migration and invasion

Nutrient starvation has been shown to be a driver of cancer cell invasion and metabolic plasticity accompanies the metastatic process [40,41]. In addition, ECM remodelling plays a pivotal role in controlling cell migration and invasion [42]. Therefore, we investigated whether PAK1-mediated ECM uptake and tyrosine catabolism by HPDL were required for breast cancer cell migration on CAF-generated CDM under amino acid starvation. Mesenchymal cancer cells, such as MDA-MB-231 cells, migrate on CDM by elongating long pseudopods in the direction of cell migration (**Fig 8A**), and this process correlates with cancer cell invasive

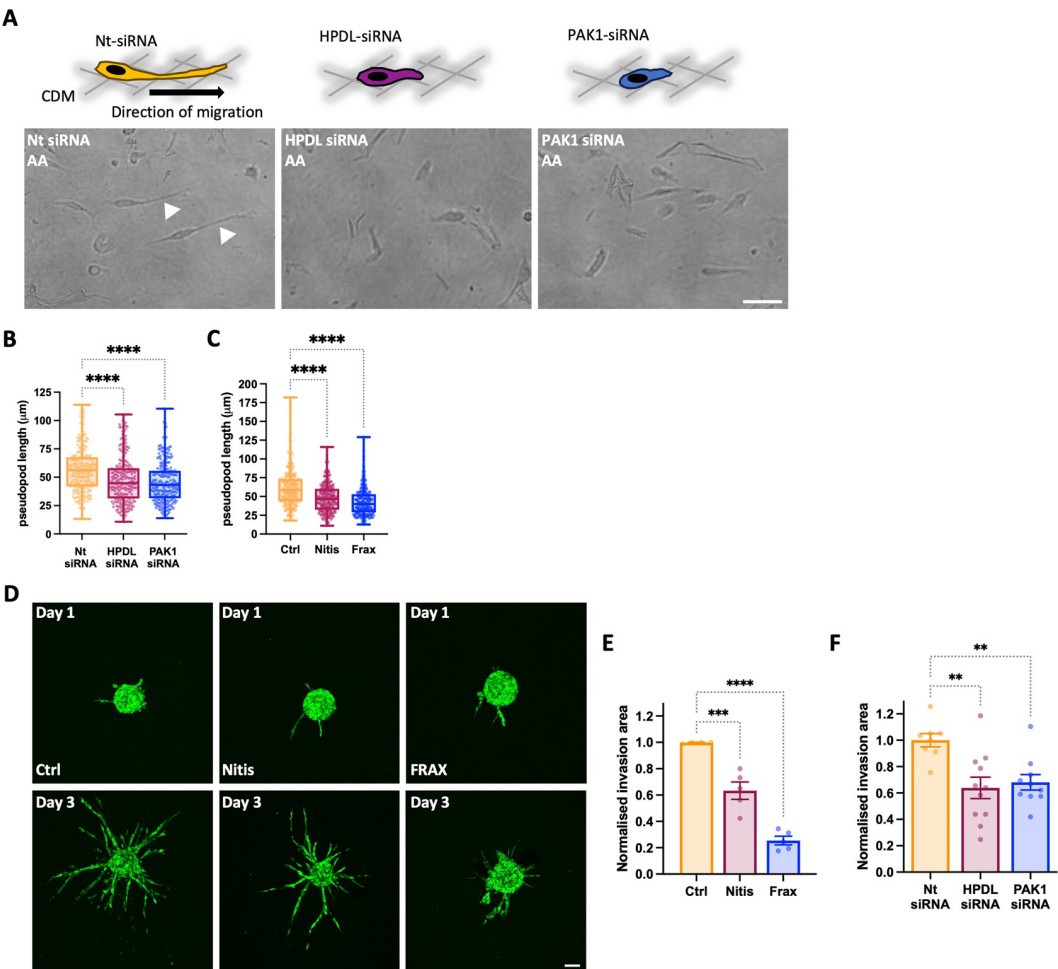

**Fig 8. Tyrosine catabolism supported cell migration and invasion in 3D systems.** (A–C) MDA-MB-231 cells, transfected with siRNA targeting PAK1 (PAK1 siRNA), HPDL (HPDL siRNA), or a non-targeting siRNA control (Nt siRNA) when indicated, where plated on CAF-CDM for at least 4 h, treated with 3 μm FRAX597 (FRAX) or 40 μm Nitisinone (Nitis) and imaged for 16 h by time lapse microscopy. Arrowheads indicate elongated pseudopods. Bar, 30 μm. Pseudopod elongation was measured with Image J. Values are from 3 independent experiments, ****$p < 0.0001$ Kruskal–Wallis, Dunn's multiple comparisons test. (D–F) GFP-MDA-MB-231 cell spheroids were generated by the hanging drop method, embedded in 3 mg/ml collagen I and geltrex (50:50) mixture. Cells were transfected or spheroids were treated as in A–C, starved of amino acid and imaged live with a Nikon A1 confocal microscope. Invasion area was calculated with Image J, by subtracting the core area from the total spheroid area. Values are mean ± SEM from 3 independent experiments, **$p < 0.01$; ***$p < 0.001$; ****$p < 0.0001$ Kruskal–Wallis, Dunn's multiple comparisons test. All the raw data associated with this figure are available in S9 Data. CAF, cancer-associated fibroblast; CDM, cell-derived matrix; HPDL, p-hydroxyphenylpyruvate hydroxylase-like protein.

abilities [11,43]. Interestingly, both siRNA-mediated down-regulation (**Fig 8B**) and pharmacological inhibition (**Fig 8C**) of PAK1 and HPDL significantly reduced the pseudopod length of migrating cells. Consistently, PAK1 and HPDL inhibition significantly impaired MDA-MB-231 spheroid invasion under amino acid starvation (**Fig 8D–8F**). Together, these data indicate that ECM-dependent tyrosine metabolism also impinges on cancer cell migration and invasion through complex 3D matrices.

## Discussion

Breast cancer cells are embedded into a highly fibrotic microenvironment enriched in several ECM components [3]. Different studies suggested that cancer cells rely on extracellular protein internalisation during starvation [7,9–12]. However, to date no studies investigated the

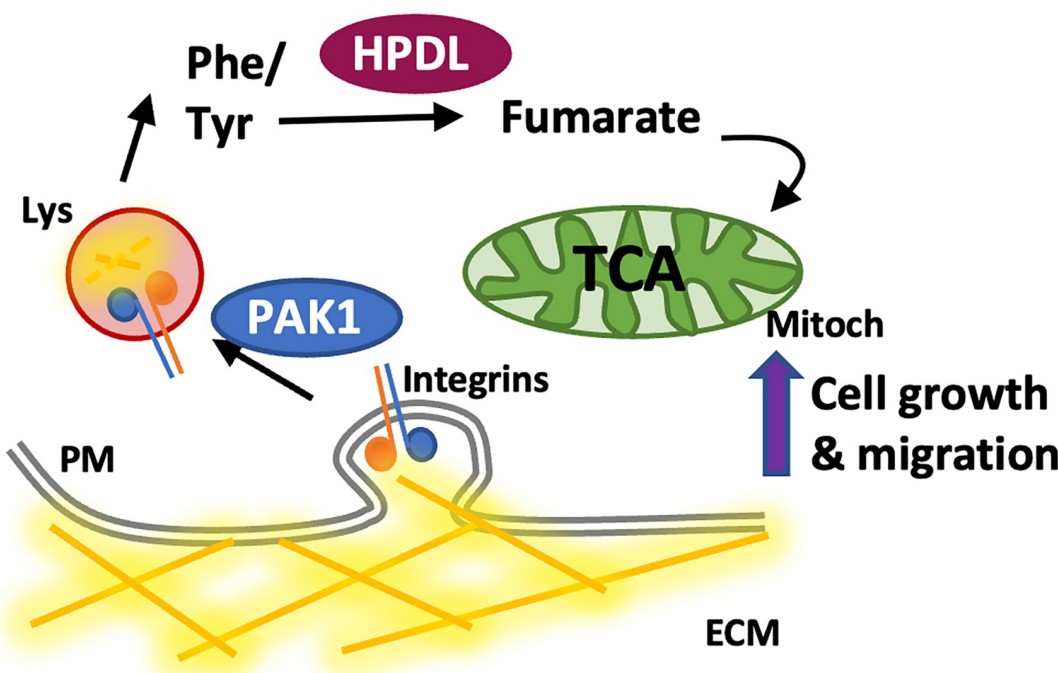

**Fig 9. Working model.** Invasive breast cancer cells internalised ECM components through PAK1-mediated macropinocytosis, resulting in ECM lysosomal degradation. This up-regulated phenylalanine/tyrosine catabolism, leading to an HPDL-dependent increased production of fumarate, supporting cell proliferation and invasive cell migration under amino acid starvation. ECM, extracellular matrix; HPDL, p-hydroxyphenylpyruvate hydroxylase-like protein; PM, plasma membrane; Lys, lysosome; Mitoc, mitochondria.

behaviour of invasive breast cancer cells in response to the presence of ECM under nutrient starvation. Here, we demonstrated that ECM internalisation and lysosomal degradation provide a source of amino acids, which supports cell growth and invasive cell migration in a phenylalanine and tyrosine metabolism-dependent manner, when invasive breast cancer cells experience amino acid starvation (**Fig 9**). Importantly, we showed that the ability to use ECM to sustain cell growth under starvation was progressively acquired during carcinoma progression, as the ECM was unable to rescue cell growth in non-transformed MCF10A; while Matrigel, but not collagen I, promoted the growth of noninvasive MCF10A-DCIS cells under glutamine, but not amino acid, deprivation. Noninvasive breast cancer cells reside inside the mammary duct and are in contact with the BM, but not with stromal collagen I [44]. Since Matrigel is a BM extract, our data suggest that MCF10A-DCIS cells are only able to take advantage of BM components to support their growth under starvation. Consistent with our data in MDA-MB-231 cells, highly invasive and metastatic MCF10CA1 cell growth was supported by the ECM under both starvation conditions. This is consistent with the fact that ECM internalisation is strongly up-regulated in invasive breast cancer cells compared to noninvasive MCF10A-DCIS and non-transformed MCF10A cells. In contrast, it was previously demonstrated that the growth of serum starved MCF10A cells was promoted by the incubation with soluble laminin [9], suggesting that the role of the ECM in supporting non-transformed epithelial cell growth could be dependent on the type of starvation experienced by the cells. Indeed, we observed a partial rescue of MCF10A cell growth in the presence of Matrigel under serum starvation (**S9 Fig**).

We found that ECM generated by CAFs, but not NFs, was able to rescue breast cancer cell growth under amino acid starvation. Similarly, it has recently been published that mouse

CAF-CDM, but not NF-CDM promotes the growth of breast and pancreatic cancer cells [17]. These matrices have been shown to have different stiffnesses (0.4 kPa for NF-CDM versus 0.8 kPa for CAF-CDM [45]), raising the intriguing possibility that changes in stiffness might promote cell growth under starvation. Stiffness has been shown to control metabolism in endothelial cells, where it stimulated glutaminolysis [46]. However, our results were obtained under amino acid starvation, therefore in the absence of glutamine, and glutaminolysis was not identified among the ECM-enriched pathways under amino acid starvation. In addition, we did not detect significant changes in ECM endocytosis and cell proliferation under amino acid starvation in response to different substrate rigidity, suggesting that stiffness does not play a prominent role in the regulation of ECM uptake and ECM-dependent cell growth under starvation. Moreover, we detected minor changes in cell proliferation in complete media, when comparing plastic and ECM, and on gels of different stiffnesses. Although it is widely established in the literature that soft ECMs result in impaired cell proliferation, our experiments were performed in the presence of dialysed FBS (DFBS), which might dampen cell proliferation on plastic.

It has recently been reported that amino acid starvation induced MT1-MMP-dependent ECM degradation, through the inhibition of MT1-MMP endocytosis [47]. Here, we showed that MMP inhibition did not oppose ECM internalisation and ECM-dependent cell growth. However, Colombero and colleagues assessed ECM degradation within the first 24 h after amino acid starvation [47], while we focused on later time points (3 and 6 days). Therefore, it is possible that the induction of ECM degradation is an early response to amino acid starvation, which is not required to sustain cell proliferation at later time points. In addition, collagen I endocytosis was recently shown not to be affected by GM6001 treatment in lung adenocarcinoma cells [48].

We showed that the presence of ECM fully rescued mTORC1 activation under amino acid starvation. This is in agreement with several previous studies showing that mTORC1 could be reactivated by extracellular proteins under amino acid deficiency [9,10] through a positive feedback loop, as inhibition of mTORC1 induced extracellular protein scavenging [9,10]. Similarly, we have previously demonstrated that mTORC1 inhibition promoted fibronectin internalisation in ovarian cancer cells [11]. Furthermore, our data showed that the presence of ECM elevated mTORC1 activity in starved cells to the same level as cells in complete media; whereas, the ECM only partially rescued cell growth under starvation, indicating that growth is still limited despite a complete activation of mTORC1. It is important to note that we used only 1 readout of mTORC1 activation (S6 phosphorylation); therefore, we cannot exclude the possibility that the ECM was unable to rescue other signalling pathways downstream of mTORC1. Interestingly, it has been previously shown that elevated mTORC1 activity restrains growth if cells rely on scavenging extracellular proteins as a source of amino acids. This is because high mTORC1 activity results in higher translation rate and cells relying on extracellular proteins cannot support the high rate of both protein translation and growth at the same time, due to their reduced access to free amino acids. Therefore, mTORC1 reactivation resulted in lower growth in cells using extracellular proteins as amino acid source [49]. Consistently, the reactivation of mTORC1 by MHY1485 treatment did not increase cell proliferation in MCF10A and MCF10A-DCIS cells. Previous literature showed that, in pancreatic cancer cells, collagen-dependent cell growth under glucose and glutamine starvation was coupled with activation of ERK, but not mTORC1, signalling [13], suggesting that the ECM supports cell growth via triggering distinct downstream signalling pathways in different cancer types.

We characterised the endocytic pathways controlling the internalisation of different ECM components and, consistently with previous observation [26,48], we found that different pathways contributed to ECM uptake, including macropinocytosis, whose inhibition resulted in

the strongest reduction in CDM and collagen I uptake. PAK1 is a well-established regulator of macropinocytosis [29], and it has been shown to be up-regulated in a variety of cancer types, including breast cancer [50,51]. PAK1 can be activated downstream of multiple signalling pathways, including the small GTPases Rac and Cdc42 as well as Phosphatidyl inositol 3 kinase (PI3K)/AKT. By using both matrix cross-linking and PAK1 inhibition, we demonstrated that ECM internalisation is required for ECM-dependent cell growth under amino acid starvation.

Phenylalanine and tyrosine metabolism has been shown to be implicated in several cancer types. Interestingly, a comprehensive metabolomic study identified the up-regulation of phenylalanine and tyrosine catabolism in invasive ductal carcinoma compared to benign tumours and normal mammary tissues [52]. HPD has been shown to be up-regulated in invasive breast cancer and, to a lesser extent, DCIS compared to normal tissues and its expression correlated with poor prognosis [53]. In lung cancer, high HPD expression correlated with poor prognosis. HPD has been reported to promote cell growth in vitro and in vivo, by controlling the flux through the pentose phosphate pathway, in a tyrosine metabolism-independent manner [37]. It is therefore possible that HPD controls cell growth on plastic via a similar mechanism in our system as well. In colon adenocarcinoma cells, it has been shown that HPDL knock-down reduced oxygen consumption, implying that it could play a role in controlling mitochondria functions. While we could detect a reduction in fumarate levels upon HPDL knock-down, in colon cancer cells HPDL knock-down did not result in impaired tyrosine catabolism [38], suggesting that, similarly to HPD, HPDL could be involved in multiple metabolic pathways. Interestingly, breast cancer patient survival analysis indicated that high expression of HPDL correlated with a reduced survival. Future work will characterise the mechanism through which HPDL controls ECM-dependent cell proliferation. Our data suggest that ECM-derived fumarate, generated through tyrosine catabolism, fuels the TCA cycle, as we could detect the accumulation of other TCA intermediates, including $a$-ketoglutarate, succinate, and malate in cells growing on ECM under amino acid starvation (**S10 Fig**). In agreement with this, it has been previously reported that, in the presence of collagen I, a significant proportion of the TCA intermediate citrate is not labelled by glucose or glutamine, suggesting that the collagen matrix could represent a fuel source in the TCA cycle [54]. However, fumarate has also been proposed to behave as an oncometabolite, and its accumulation has been shown to promote the stabilisation of hypoxia-inducible factor (HIF) [55]. Further work will elucidate the molecular mechanisms through which the increase in fumarate levels generated by the activation of tyrosine catabolism supports ECM-dependent cell growth under amino acid starvation.

Nutrient starvation, including amino acid limitation, has been shown to drive cancer cells invasion and metastasis, through the activation of the integrated stress response pathway and translational reprogramming [40,41]. For instance, in pancreatic cancer cells, glutamine deprivation promotes invasion and metastasis through the activation of a Slug-dependent epithelial-to-mesenchymal transition programme [56]. In migrating cells, the ECM has been shown to promote a shift from glycolysis to oxidative phosphorylation [57], raising the intriguing hypothesis that ECM internalisation and the subsequent increase in the intracellular amino acid pool could facilitate mitochondrial ATP production, required to sustain cell migration and metastasis. Indeed, we observed that the inhibition of ECM endocytosis by PAK1 knock-down and the blockade of tyrosine catabolism by HPDL down-regulation opposed cancer cell migration and invasion in 3D system. PAK1 is a key downstream effector of the small GTPase Rac1, which plays important roles in cancer cell migration and invasion [58]. Importantly, mesenchymal cell migration on CDM has been shown to be driven by the inhibition of Rac and the activation of RhoA signalling [59], suggesting that the reduced cell migration we observed in the presence of PAK inhibition is not due to a reduction in Rac signalling. In addition, fumarate could also play a more direct role on cell migration, as it has been reported to

activate epithelial-to-mesenchymal transition in renal cell carcinoma [60]. Future work will dwell into the role of ECM endocytosis in cancer cell migration and invasion. A limitation of our study is that we were not able to detect differences in tumour growth and lung metastasis in an orthotopic model of breast cancer, driven by the Polyoma Middle T (PyMT) oncogene. It is important to consider that the PyMT model transcriptionally resembles Luminal B breast cancer, while in human tumours HPDL overexpression is mainly observed in triple negative basal-like breast cancers (S11 Fig), suggesting that the role of tyrosine catabolism might be subtype specific. Further work will characterise the subtype specificity of ECM-dependent cell growth both in vitro and in vivo.

In conclusion, our findings demonstrated that, under amino acid starvation, invasive breast cancer cells, but not non-transformed mammary epithelial cells and noninvasive DCIS cells, internalised and degraded ECM components. This resulted in an increase in tyrosine catabolism, leading to the production of fumarate. Therefore, HPDL-mediated tyrosine catabolism could represent a metabolic vulnerability of cancer cells thriving in a nutrient-deprived microenvironment.

## Methods

### Reagents

Primary antibodies for Phospho-S6 Ribosomal Protein ser235/236, mTOR, phospho-4EBP1, and GAPDH were from Cell Signaling, Phospho-FAK (Tyr397), HPDL from Thermo Fisher, Paxilin and LAMP2 from BD-bioscience; PAK1 from Proteintech. Secondary antibodies Alexa-fluor 594 anti-Rabbit IgG, Alexa-fluor 488 anti-Rabbit IgG, Alexa-fluor 594 anti-Mouse IgG and Alexa-fluor 488 anti-Mouse IgG were from Cell Signaling, IRDye 800CW and IRDye 680CW were from LI-COR. Alexa fluor TM 555 Phalloidin, Click-iT EdU Imaging Kits, CellEvent Caspase-3/7 Green Detection kit, NHS-Fluorescein, NHS-Alexa Fluor 555, pH-rodo iFL STP ester red and Hoechst 33342 were from Invitrogen. DRAQ5 was from LI-COR. Collagen I and Matrigel were from Corning. Geltrex was from Thermo Fisher. All media and dialyzed FBS were from Gibco, except for DMEM with no amino acid which was from US Biological life science and Plasmax which was kindly provided by Dr Tardito, CRUK Scotland Institute, Glasgow. The details of Plasmax composition were previously described by Vande Voorde and colleagues [18]. E64d (Aloxistatin) and PF573228 were from AdooQ Bioscience. GM6001 was from APEXBIO. Dynasore, Filipin, and EIPA were from Sigma. FRAX 597 was from bio-Techne. Nitisinone was from MCE. PP2 was from Generon. C13-tyrosine was from Sigma-Aldrich.

### Cell culture

MDA-MB-231 cells, Telomerase immortalised NFs and CAFs, generated in Professor Akira Orimo's lab, Paterson Institute, Manchester, were cultured in High glucose Dulbecco's Modified Eagle's Medium (DMEM) supplemented with 10% foetal bovine serum (FBS) and 1% penicillin streptomycin (PS). MCF10A, MCF10A-DCIS, and MCF10CA1 were a kind gift from Prof Giorgio Scita, IFOM, Milan (Italy). MCF10A-DCIS cells were cultured in DMEM/F12 supplemented with 5% Horse serum (HS), 20 ng/ml epithelial growth factor (EGF), and 1% PS. MCF10A cells were cultured in DMEM/F12 supplemented with 5% HS, 20 ng/ml EGF, 0.5 mg/ml hydrocortisone, 10 μg/ml insulin and 1% PS. MCF10CA1 cells were cultured in DMEM/F12 supplemented with 2.5% HS, 20 ng/ml EGF, 0.2 mg/ml hydrocortisone, 10 μg/ml insulin, and 1% PS. Cells were grown at 5% $CO_2$ and 37˚C and passaged every 3 to 4 days. All cell lines were routinely tested for mycoplasma contamination.

## ECM preparation

To coat plates/dishes, collagen I and Matrigel were diluted with cold PBS on ice to 2 mg/ml and 3 mg/ml concentrations, respectively, and incubated for 3 h at 5% $CO_2$ and 37˚C for polymerisation. CDM was generated from NFs and CAFs as described in [14]. Briefly, plates were coated with 0.2% gelatin in PBS, incubated at 37˚C for 1 h and cross-linked with 1% glutaraldehyde in PBS at room temperature (RT) for 30 min. Glutaraldehyde was removed and quenched in 1 M glycine at RT for 20 min, followed by 2 PBS washes, and an incubation in complete media at 5% $CO_2$ and 37˚C for 30 min. CAFs or NFs were seeded and incubated at 5% $CO_2$, 37˚C until fully confluent. The media was replaced with full growth media containing 50 μg/ml ascorbic acid, and it was refreshed every 2 days. NFs were kept in media supplemented with ascorbic acid for 6 days while CAFs for 7 days. The media was aspirated, and the cells washed once with PBS containing calcium and magnesium. The cells were incubated with extraction buffer containing 20 mM $NH_4OH$ and 0.5% Triton X-100 in PBS with calcium and magnesium for 2 min at RT until no intact cells were visible by phase contrast microscopy. Residual DNA was digested with 10 μg/ml DNase I at 5% $CO_2$, 37˚C for 1 h. The CDMs were kept in PBS containing calcium and magnesium at 4˚C.

## ECM cross-linking

The ECMs were prepared as described above and treated with 10% glutaraldehyde for 30 min at RT, followed by 2 PBS washes. The cross-linker was quenched in 1 M glycine at RT for 20 mins, followed by 2 PBS washes. Cross-linked ECMs were kept in PBS at 5% $CO_2$ and 37˚C overnight.

For ribose cross-linking, glass-bottom dishes were coated with 1 mg/ml collagen I (diluted in 100 mM $NaHCO_3$ in PBS) and let polymerise for 1 h at 37˚C. Collagen I-coated dishes were then incubated with 200 mM ribose or the vehicle PBS for 48 h at 37˚C, as previously described in [36,61].

## Starvation conditions

The media used for the starvations were supplemented with either 10% dialyzed FBS (DFBS, for MDA-MB-231 cells); 10% HS, 10 μg/ml insulin and 20 ng/ml EGF (for MCF10A cells); 5% HS and 20 ng/ml EGF (for MCF10A-DCIS cells) or 2.5% HS, 20 ng/ml EGF, 0.2 mg/ml hydrocortisone and 10 μg/ml insulin (for MCF10CA1 cells). Cells were plated in complete media for 5 h, then the media was replaced with the starvation media as indicated in Table 1. The protein content of FBS, DFBS, and HS was analysed by mass spectrometry. Tables 2 and 3 contain the serum components identified as ECM or ECM-associated proteins in the matrisome database (https://matrisomedb.org/).

## Proliferation assays

The 96-well plates were coated with 15 μl/well of 2 mg/ml collagen I or 3 mg/ml Matrigel. Three wells of 96-well plate were allocated to each condition as technical replicates, and $10^3$

**Table 1. Starvation media.**

| Media | Supplier | Ref No |
|---|---|---|
| DMEM, high glucose, no glutamine | Gibco | 11960–044 |
| DMEM, no amino acids | Stratech | D9800-13-USB-10L |
| DMEM/F-12, no glutamine | Gibco | 21331–020 |

**Table 2. ECM proteins detected in horse serum.**

| Name | Gene | Intensity |
|---|---|---|
| Alpha-2-macroglobulin | A2M | 1.58E+09 |
| Adiponectin D | ADIPOQ | 1.61E+08 |
| Angiotensinogen | AGT | 1.99E+09 |
| Protein AMBP | AMBP | 1.18E+09 |
| Adiponectin A | C1QB | 1.92E+08 |
| Complement C1q C chain | C1QC | 2.73E+08 |
| Carboxypeptidase N subunit 2 | CPN2 | 2.17E+08 |
| Extracellular matrix protein 1 | ECM1 | 5.41E+08 |
| Coagulation factor XII | F12 | 4.53E+08 |
| Prothrombin | F2 | 2.20E+09 |
| Fibulin-1 | FBLN1 | 8.13E+07 |
| Fibrinogen alpha chain | FGA | 4.49E+08 |
| Fibrinogen gamma chain | FGG | 2.00E+07 |
| Fibronectin | FN1 | 6.32E+09 |
| Hemopexin | HPX | 3.69E+09 |
| Histidine-Rich glycoprotein | HRG | 1.16E+10 |
| Insulin like growth factor binding protein acid labile subunit | IGFALS | 1.23E+08 |
| Inter-alpha-trypsin inhibitor heavy chain H1 | ITIH1 | 5.60E+09 |
| Inter-alpha-trypsin inhibitor heavy chain H2 | ITIH2 | 6.68E+09 |
| Inter-alpha-trypsin inhibitor heavy chain 3 | ITIH3 | 6.81E+08 |
| Inter-alpha-trypsin inhibitor heavy chain 4 | ITIH4 | 5.01E+09 |
| Kininogen 1 | KNG1 | 1.82E+09 |
| Leucine rich alpha-2-glycoprotein 1 | LRG1 | 2.17E+08 |
| Lumican | LUM | 3.10E+07 |
| Plasminogen | PLG | 3.07E+09 |
| Serpin family A member 10 | SERPINA10 | 3.22E+08 |
| Serpin family A member 3 | SERPINA3 | 8.73E+08 |
| Serpin family A member 6 | SERPINA6 | 7.13E+07 |
| Serpin family A member 7 | SERPINA7 | 9.80E+07 |
| Antithrombin-III | SERPINC1 | 1.82E+10 |
| Serpin family D member 1 | SERPIND1 | 2.10E+09 |
| Serpin family F member 2 | SERPINF2 | 9.49E+08 |
| Transforming growth factor beta induced | TGFBI | 3.01E+07 |
| Vitronectin | VTN | 3.70E+08 |

cells/well (MDA-MB-231, MCF10A, MCF10A-DCIS cells) or 400 cells/well (MCF10CA1) were seeded in complete growth media. After a 5-h incubation in 5% $CO_2$ and 37°C, the media was replaced with 200 μl of the starvation media, in the presence of 10 μm GM6001, 0.5 μm PF573228, 2.5 or 5 μm PP2, 3 μm FRAX597, 40 μm Nitisinone or DMSO control where indicated. The inhibitors were supplemented every 2 days, up to day 4 or 6. Cells were fixed by adding 4% paraformaldehyde (PFA) for 15 min at RT followed by 2 PBS washes. Cells were stained with either DRAQ5 (assays on collagen I and Matrigel) or Hoechst 33342 (assays on CDM or cross-linked ECM, where the thickness and structure of the matrices resulted in a very high background and accurate readings could not be obtained with DRAQ5), and 5 μm DRAQ5 in PBS was added to the cells for 1 h at RT with gentle rocking. Cells were washed twice with PBS for 30 min to avoid background fluorescence and kept in PBS for imaging. DRAQ5 was detected by the 700 nm channel with 200 μm resolution with an Odyssey Sa

**Table 3. ECM proteins detected FBS and DFBS. ND = not detected.**

| Name | Gene | Intensity FBS | Intensity DFBS |
|---|---|---|---|
| C-type lectin domain containing 11A | CLEC11A | ND | 2.61E+07 |
| Collagen type X alpha 1 chain | COL10A1 | ND | 2.69E+07 |
| Collagen type I alpha 2 chain | COL1A2 | 1.01E+08 | 6.81E+07 |
| Collagen type V alpha 1 chain | COL5A1 | 5.57E+07 | 2.92E+07 |
| Collagen type VI alpha 1 chain | COL6A1 | 4.12E+08 | 2.17E+08 |
| Collagen type VI alpha 3 chain | COL6A3 | 1.09E+08 | 4.95E+07 |
| ECM1 protein | ECM1 | 1.46E+07 | ND |
| EGF containing fibulin extracellular matrix protein 1 | EFEMP1 | 4.30E+07 | ND |
| Coagulation factor XIII A chain | F13A1 | 6.29E+07 | 2.74E+07 |
| Fibrinogen gamma chain | FGG | 4.55E+08 | 2.66E+08 |
| Fibromodulin | FMOD | 4.18E+07 | ND |
| Fibronectin | FN1 | 1.68E+09 | 8.08E+08 |
| Hyaluronan-binding protein 2 | HABP2 | 4.71E+07 | ND |
| MBL associated serine protease 2 | MASP2 | 3.23E+07 | 1.38E+07 |
| OGN protein | OGN | 3.36E+07 | ND |
| Procollagen C-endopeptidase enhancer | PCOLCE | 4.18E+07 | 1.13E+07 |
| SERPINA11 protein | SERPINA11 | 2.04E+07 | ND |
| SPARC | SPARC | 4.84E+07 | ND |
| Transforming growth factor beta induced | TGFBI | 6.86E+07 | 4.74E+07 |
| Tenascin XB | TNXB | 2.20E+07 | ND |

instrument. The signal intensity (total intensity minus total background) of each well was quantified with Image Studio Lite software. Alternatively, cells were fixed with 4% PFA containing 10 μg/ml Hoechst 33342 for 15 min, followed by 2 PBS washes. Images were acquired by ImageXpress micro with a 2× objective and the whole area of each well was covered. The images were analysed with MetaXpress and Costume Module Editor (CME) software in the Sheffield RNAi Screening Facility (SRSF). CAF/NF-CDM was generated in 96-well plates and cells were seeded as described above. Cells were fixed with 4% PFA containing 10 μg/ml Hoechst 33342 for 15 min, washed twice with PBS, permeabilised with 0.25% Triton X-100 for 5 min, and stained with Phalloidin Alexa Fluor 488 (1:400 in PBS) for 30 min. Cells were washed twice with PBS and were left in PBS for imaging. Images were collected by ImageXpress micro and analysed by MetaXpress and CME software.

## EdU incorporation assays

The 96-well plates were coated with ECM and $10^3$ MDA-MB-231 cells/well were seeded in full growth media. After a 5-h incubation at 5% $CO_2$ and 37˚C, the full growth media was replaced with 200 μl of the starvation media. At day 6 post starvation, cells were incubated with 5 μm EdU for 2 days at 5% $CO_2$ and 37˚C, fixed with 4% PFA containing 10 μg/ml Hoechst 33342 for 15 min at RT and permeabilised with 0.25% Triton X-100 for 5 min. Cells were incubated with EdU detection cocktail (Invitrogen, Click-iT EdU Alexa Fluor 555) for 30 min at RT with gentle rocking. Cells were washed twice with PBS and were kept in PBS for imaging. Images were collected by ImageXpress micro with a 2× objective and quantified with MetaXpress and CME software.

## Caspase-3/7 detection

The 96-well plates were coated with ECM and $10^3$ MDA-MB-231 or MCF10CA1 cells/well were seeded in full growth media. After a 5-h incubation at 5% $CO_2$ and 37˚C, the full growth

media was replaced with 200 μl of the starvation media. Cells were kept under starvation for 3 or 6/8 days, then the media were replaced by 5 μm CellEvent Caspase-3/7 Green Detection Reagent diluted in PBS, containing calcium and magnesium, and 5% DFBS for 1.5 h at 5% $CO_2$ and 37˚C. Cells were fixed with 4% PFA containing 10 μg/ml Hoechst 33342 for 15 min at RT, washed and left in PBS for imaging. Images were collected by ImageXpress micro with a 10× objective and analysed with MetaXpress and CME software.

## Propidium iodide (PI) live-cell imaging

MDA-MB-231 and MCF10CA1 cells were seeded on 96-well plates coated with ECM under full growth media. After a 5-h incubation at 5% $CO_2$ and 37˚C, the full growth media was replaced with the starvation media. After 3, 6, or 8 days of starvation, cells were incubated with 1 μg/ml PI and 5 μg/ml Hoechst 3342 at 5% $CO_2$ and 37˚C for 30 min. Cells were washed once with PBS and were kept in media with no phenol red for imaging. Images were collected by ImageXpress micro with a 10× objective and quantified with MetaXpress and CME software.

## Immunofluorescence

$10^3$ MDA-MB-231 cells/well were seeded in full growth media in 96-well plates coated with ECM. After a 5-h incubation at 5% $CO_2$ and 37˚C, the full growth media was replaced with 200 μl of the starvation media. After 3 days, cells were fixed with 4% PFA containing 10 μg/ml Hoechst 33342 for 15 min at RT and permeabilised with 0.25% Triton X-100 for 5 min. Unspecific binding sites were blocked by incubating with 3% BSA at RT with gentle rocking for 1 h. Cells were incubated with anti-Phospho-S6 Ribosomal protein antibody (1:100 in PBS) overnight at 4˚C. The cells were washed twice with PBS for 5 to 10 min and incubated with secondary antibody (anti-Rabbit IgG Alexa Fluor 594, 1:1,000 in PBS) for 1 h at RT with gentle rocking. Cells were washed twice with PBS for 10 to 30 min and were kept in PBS for imaging. Images were collected by ImageXpress micro with a 10× objective and analysed with MetaXpress and CME software.

$4 \times 10^4$ MDA-MB-231 cells were seeded on ECM coated 3.5 $cm^2$ glass-bottomed dishes and incubated at 5% $CO_2$ and 37˚C for 5 or 24 h. The full growth media was then replaced with the starvation media. After 1 or 3 days, cells were fixed with 4% PFA for 15 min and permeabilised with 0.25% Triton X-100 for 10 min, followed by 1 h incubation in 3% BSA (blocking buffer). Cells were incubated with anti-Paxillin (1:200 in PBS), anti-HPDL (1:100 in blocking buffer), anti-mTOR (1:100 in blocking buffer), or anti-LAMP2 (1:100 in blocking buffer) primary antibodies at RT for 1 h. Cells were washed 3 times with PBS for 20 min and incubated with Alexa-Fluor 488 anti-Mouse IgG, Alexa-Fluor 594 anti-Mouse IgG, or Alexa-Fluor 488 anti-rabbit IgG secondary antibodies (1:1,000 in blocking buffer) for 1 h at RT with gentle shaking and protected from light, and 2 to 3 drops of Vectashield mounting medium containing DAPI were added to the dishes, which were then sealed with parafilm and kept at 4˚C. Cells were visualised by confocal microscopy using a Nikon A1 microscope and 60×, 1.4 NA oil immersion objective. The FA index was calculated by thresholding the paxillin signal, to remove the cytoplasmic staining and only retain paxillin signal from FAs. The area of paxillin-positive objects in a cell was normalised to the total area of the cell. Similarly, mTOR endosomal index was calculated dividing the area covered by mTOR positive vesicles (obtained by thresholding the mTOR signal) by the total cell area. mTOR/LAMP2 co-localisation (Icorr index) was calculated using the co-localisation analysis plug in developed by [62].

## ECM uptake

3.5 $cm^2$ glass-bottomed dishes were coated with 1 mg/ml or 2 mg/ml collagen I or 3 mg/ml Matrigel and incubated at 5% $CO_2$ and 37˚C for 3 h. ECMs were cross-linked where indicated

and labelled with 10 µg/ml Fluorescein or Alexa Fluor 555 succinimidyl esters (NHS) for 1 h at RT on a gentle rocker. $10^5$ MDA-MB-231 cells per dish were seeded in full growth media. After a 5-h incubation in 5% $CO_2$ and 37˚C, full growth media were replaced with 1 ml of the indicated starvation media in the presence of 20 µm E64d or DMSO. E64d and DMSO were added after 2 days and cells were fixed at day 3 by adding 4% PFA for 15 min at RT. Cells were permeabilised with 0.25% Triton X-100 for 5 min and incubated with Phalloidin Alexa Fluor 555 (1:400 in PBS) for 10 min, and 2 to 3 drops of Vectashield mounting medium containing DAPI were added to the dishes, which were then sealed with parafilm and kept at 4˚C. Cells were visualised by confocal microscopy using a Nikon A1 microscope and 60× 1.4 NA oil immersion objective. Alternatively, collagen I or CDMs were labelled with 10 µg/ml pH-rodo-red iFL for 1 h at RT. Dishes were washed twice before cell seeding; $3 \times 10^5$ MDA-MB-231, MCF10A, MCF10A-DCIS, and MCF10CA1 cells were incubated for 6 h and stained with Hoechst 33342 for nuclear labelling prior to live cell imaging with a Nikon A1 microscope and 60× 1.4 NA oil immersion objective.

For the high-throughput imaging of ECM endocytosis, $10^4$ MDA-MB-231 cells/well were seeded on pH-rodo labelled 0.5 mg/ml Matrigel, 0.5 mg/ml collagen I or NF-CDM in 384-well plates in the presence of DMSO control, 20 µm or 40 µm Dynasore, 3.75 µg/ml or 5 µg/ml filipin, 25 µm or 35 µm EIPA for 6 h, labelled with 10 µg/ml Hoechst 33342 and imaged live with a 63× water immersion objective with an Opera Phenix microscope. 2.5 µl of 500 nM siGENOME siRNA smart pool and 2.5 µl Opti-MEM per well was added into CellCarrier Ultra 384 well plates (Perkin Elmer); 4.95 µl Opti-MEM was incubated with 0.05 µl Dharmafect IV for 5 min; 5 µl of the Dharmafect IV solution was added into each well. Plates were incubated for 20 min on gentle rocker at RT; $3 \times 10^3$ cells were seeded in 40 µl DMEM containing 10% FBS. The final concentration of the siRNA was 25 nM. Cells were kept at 5% $CO_2$ and 37˚C for 72 h. Transfected cells were transferred into pH-rodo-labelled 0.5 mg/ml Matrigel-coated CellCarrier Ultra 384 well plates. Cells were incubated for 6 h, stained with Hoechst 33342, and imaged live with a 40× water immersion objective with Opera Phenix microscope. Images were analysed with Columbus software. ECM uptake was quantified with Fiji [63] as described in [64].

## Polyacrylamide hydrogel (PAAG) preparation

Glass-bottom dishes (20 mm glass diameter) were treated with 1 M NaOH for 1 h at RT. The solution was aspirated, dishes were washed twice with distilled water and twice with ethanol. Dishes were incubated with Blind-Silane solution for 1 h at RT, followed by 3 washes with Ethanol; 22 µl of gels with different stiffness (Table 4) were added to the glass part of the dish and covered with 18 mm coverslips for 1 h at RT to solidify. After solidification, PBS was added to each dish for 30 min. Coverslips were removed and PAAGs were washed twice with PBS. To functionalise the PAAG, they were incubated with 100 µl of Sulfo-SANPAH under a UV lamp for 5 min. PAAGs were washed 4 times with PBS with gentle rocking and 3 times with 50 mM HEPES. The PAAGs were coated with 200 µl of 1 mg/ml collagen I overnight at 5% $CO_2$ and

**Table 4. PAAG composition in µl.**

| Reagents | 0.7 kPa | 3 kPa | 6 kPa |
|---|---|---|---|
| PBS | 436.25 | 402.67 | 385.03 |
| 40% acryl | 50 | 68.6 | 93.72 |
| 2%bis acryl | 7.5 | 22.48 | 15 |
| APS | 2.5 | 2.5 | 2.5 |
| TEMED | 0.25 | 0.25 | 0.25 |

37˚C. The day after, collagen I was washed twice with PBS and incubated in full growth media for 1 h at 5% $CO_2$ and 37˚C. The media was removed and PAAGs were kept inside the hood for 1 to 2 min. Then, $5 \times 10^4$ cells/dish were seeded in full growth media. After 5 h, full growth media was replaced with starvation media. After 5 days of starvation, cells were incubated with 10 μm EdU for 1 day at 5% $CO_2$ and 37˚C, fixed with 4% PFA for 15 min at RT and permeabilised with 0.25% Triton X-100 for 5 min. Cells were incubated with Click-iT EdU Alexa Fluor 555 (Invitrogen) for 30 min at RT with gentle rocking. Cells were washed twice with PBS and were covered by Vectashield with DAPI. Images were collected by a Nikon A1 confocal microscope.

## Western blotting

For phospho-FAK detection, MDA-MB-231 cells were trypsinised and $10^6$ cells were collected in each Falcon tube. Cells were treated with 0.2, 0.5, 0.7, and 1 μm P573228 for 30 min at 5% $CO_2$ and 37˚C, while they were kept in suspension. Cells were transferred to a 6-well plate which had been coated with 900 μl of 0.5 mg/ml collagen I. Cells were allowed to adhere for 30 min at 5% $CO_2$ and 37˚C. The media was aspirated, and cells were put on ice. After 2 washes with ice-cold PBS, 100 μl per well of lysis buffer (50 mM Tris (pH 7) and 1% SDS) were added. Lysates were collected and transferred to QiaShredder columns, which were spun for 5 min at 6,000 rpm. The filter was discarded, and the extracted proteins were loaded into 12% acrylamide gels and run at 100 V for 2.5 h. Proteins were transferred to FL-PVDF membranes in transfer buffer for 75 min at 100 V. Membranes were washed 3 times with TBS-T (Tris-Buffered Saline, 0.1% Tween) and blocked in 5% w/v skimmed milk powder in TBS-T for 1 h at RT. Membranes were then incubated with primary antibodies, 1:500 GAPDH, and 1:1,000 pFAK in TBS-T with 5% w/v skimmed milk overnight at 4˚C.

To validate the knock-down efficiency, lysates were run on 4% to 15% Mini Protean TGX stain Free Protein gels. Membranes were incubated with 1:1,000 GAPDH, *a*-tubulin and PAK1 primary antibodies in TBS-T with 5% w/v skimmed milk overnight at 4˚C. Membranes were incubated with secondary antibodies, IRDye 800CW anti-mouse IgG for GAPDH and *a*-tubulin (1:30,000) and IRDye 680CW anti-rabbit IgG for pFAK and PAK1 (1:20,000) in TBS-T with 0.01% SDS, for 1 h at RT on the rocker. Membranes were washed 3 times in TBS-T for 10 min on the rocker at RT followed by being rinsed with water. Images were taken by a Licor Odyssey Sa system. Band intensity was quantified with Image Studio Lite software.

## Non-targeted metabolite profiling

MDA-MB-231 and MCF10CA1 cells were seeded on 96- or 6-well plates coated with ECM under full growth media. After a 5-h incubation at 5% $CO_2$ and 37˚C, the full growth media was replaced with the starvation media. After 6 days, the media was removed, and cells were washed with ice-cold PBS 3 times. The PBS were aspirated very carefully after the last wash to remove every remaining drop of PBS. The extraction solution (cold; 5 MeOH: 3 AcN: 2 $H_2O$) was added for 5 min at 4˚C with low agitation. Metabolites were transferred to Eppendorf tubes and centrifuged at 4˚C at 14,000 rpm for 10 min. Samples were transferred into HPLC vials and directly injected into a Waters G2 Synapt mass spectrometer in electrospray mode within the Sheffield Faculty of Science Biological Mass Spectrometry Facility. The samples were run in positive and negative modes. Three technical replicates of each sample were run. The dataset only includes peaks present in all 3 replicates. The data is binned into 0.2 amu m/z bins and the m/z in each bin is used to identify putative IDs using the HumanCyc database. The mass spectrometry data were analysed using Perseus software version 1.5.6.0. Metabolic

pathway analysis was performed in MetaboAnalyst 5.0 (https://www.metaboanalyst.ca/, $p < 0.05$.

## Targeted metabolomics

MDA-MB-231 and MCF10CA1 cells were seeded on 6-well plates coated with ECM in full growth media. After a 5-h incubation at 5% $CO_2$ and 37˚C, full growth media was replaced with amino acid-free media. Cells were kept under starvation for 6 days; the media was removed, and metabolites were extracted as described above. Then, a mass spectrometer Waters Synapt G2-Si coupled to Waters Acquity UPLC was used to separate phenylalanine, tyrosine and fumarate with Column Waters BEH Amide $150 \times 2.1$ mm. The injection was 10 μl. The flow rate was 0.4 ml/min. The samples were run in negative mode. To identify the targeted metabolites, the retention time and the mass of the compound were matched with the standards.

## Isotope labelling

The 6-well plates were coated with 2 mg/ml collagen I or left uncoated, and $7 \times 10^4$ MDA-MB-231 cells/well were seeded in complete growth media. After a 5-h incubation in 5% $CO_2$ and 37˚C, the media was replaced with 4 ml of amino acid depleted media, in the presence of 0.4 mM C13-tyrosine for 6 days. The metabolites were extracted and analysed as described above by targeted mass spectrometry.

## Sample preparation, protein mass spectrometry, and data analysis

A total of 25 μg of protein was resuspended in 25 μl of 5% SDS containing 50 mM TEAB (pH 8.0) and digested by trypsin using S-trap technique (Protifi, United States of America) according to the manufacturer's protocol. Subsequently, peptides were dried in a vacuum concentrator before being resuspended in 60 μl of 0.5% formic acid and 1 μl was withdrawn for mass spectrometry analysis (MS). The MS analysis was performed on an Orbitrap Exploris E480 (Thermo Fisher) mass spectrometer equipped with a nanospray source, coupled to a Vanquish LC System (Thermo Fisher). Peptides were desalted online using a nano trap column, 75 μm I. D.X 20 mm (Thermo Fisher) and then separated using an EASY-Spray column, 50 cm × 50 μm ID, PepMap C18, 2 μm particles, 10 Å pore size (Thermo Fisher). The 70 min gradient was used, starting from 3% to 20% buffer B (0.5% formic acid in 80% acetonitrile) for 45 min then ramping up to 35% buffer B for 15 mins, then up to 99% buffer B for 1 min and maintaining at 99% buffer B for 9 min.

The Orbitrap Exploris was operated in positive mode with a cycle of 1 MS (in the Orbitrap) acquired at a resolution of 120,000 at m/z 400, with the top 20 most abundant multiply charged (2+ and higher) ions in a given chromatographic window subjected to MS/MS fragmentation in the linear ion trap with scan range (m/z) 375–1,200; normalised AGC target 300%; microscan 1. An FTMS target value of 1e4 and resolution of 15,000. Raw mass spectrometry data were analysed with MaxQuant version 1.6.10.43 [65]. Data were searched against Horse and Bovine Databases downloaded from UniProt (downloaded in July 2023) using the following search parameters for standard protein identification: enzyme set to Trypsin/P (2 miss-cleavages), methionine oxidation and N-terminal protein acetylation as variable modifications, cysteine carbamidomethylation as a fixed modification. A protein FDR of 0.01 and a peptide FDR of 0.01 were used for identification level cut-offs based on a decoy database searching strategy. Data from MaxQuant was exported to Excel (Microsoft, USA) for further data analysis.

**Table 5. siRNAs and sgRNAs.**

| Type | Target | Ref No |
|------|--------|--------|
| ON-TARGETplus siRNA smartpool | PAK1 | L-003521-00-0005 |
| si-GENOME siRNA smartpool | PAK1 | M-003521-04-0005 |
| CRISPRmod CRISPRi sgRNA | PAK1 | CF-003521-01-0005 |
| ON-TARGETplus siRNA smartpool | HPD | L-019648-00-0005 |
| ON-TARGETplus siRNA smartpool | HPDL | L-014985-01-0005 |
| CRISPRmod CRISPRi sgRNA | HPDL | CF-014985-01-0005 |
| ON-TARGETplus siRNA | Non-targeting siRNA 4 | D-001810-04-20 |
| siGENOME siRNA | Non-targeting siRNA 5 | D-001210-05-05 |
| CRISPRmod CRISPRi sgRNA | Non-targeting smartpool | U-009550-10-05 |
| siGENOME siRNA smartpool | Clathrin heavy chain (CHC17) | M-004001-00-0005 |
| siGENOME siRNA smartpool | Dynamin 1 | M-003940-01-0005 |
| siGENOME siRNA smartpool | Dynamin 2 | M-004007-03-0005 |
| siGENOME siRNA smartpool | Dynamin 3 | M-013931-00-0005 |
| siGENOME siRNA smartpool | Caveolin 1 | M-003467-01-0005 |
| siGENOME siRNA smartpool | Caveolin 2 | M-010958-00-0005 |
| siGENOME siRNA smartpool | Flotillin 1 | M-010636-00-0005 |
| siGENOME siRNA smartpool | Flotillin 2 | M-003666-01-0005 |
| siGENOME siRNA smartpool | GRAF 1 | M-008426-01-005 |

## Transfection

Glass-bottomed 96-well plates were coated with 15 μl 2 mg/ml collagen I and 20 μl of 150 nM Dharmacon ON TARGET plus siRNA smart pool or Dharmacon CRISPRmod CRISPRi (inhibition) synthetic sgRNA (Table 5) were added on each well; 0.24 μl of Dharmafect 1 (DF1) was diluted in 19.76 μl of DMEM and 20 μl of diluted DF1 were transferred to each well. Plates were incubated for 30 min at RT, and $3 \times 10^3$ cells in 60 $\mu$l DMEM media containing 10% FBS but no antibiotics were seeded in each well. The final concentration of siRNA was 30 nM. Cells were kept at 5% $CO_2$ and 37°C overnight. The full growth media was replaced by 200 μl of fresh complete or amino acid-depleted media for up to 6 days. At days 2 and 6, cells were fixed and incubated with 10 μg/ml Hoechst 33342. Images were collected by ImageXpress micro and analysed by MetaXpress and CME software. Alternatively, 10 μl 5 μm siRNA or sgRNA were mixed with 190 μl Opti-MEM into each well of a 6-well plate; 198 μl Opti-MEM and 2 μl DF1 were mixed and incubated for 5 min at RT. 200 μl of the Opti-MEM DF1 mix were added on top of the siRNA and incubated for 20 min on a rocker; $4 \times 10^5$ cells in 1.6 ml were added into each well. Cells were incubated at 37°C and 5% $CO_2$ for 72 h, either lysed for western blotting or seeded onto pH-rodo-labelled CDMs for 6 h and imaged live with a Nikon A1 confocal microscope, 60× 1.4 NA oil immersion objective.

## MDA-MB-231 CRISPRi cell generation

$3 \times 10^5$ cells/well were seeded into a 12-well plate. Confluent cells were transduced with CRISPRmod CRISPRi dCas9-SALL1-SDS3 lentivirus particles, which expression is under the hCMV promoter (cat VCAS10124), as per protocol (https://horizondiscovery.com/-/media/Files/Horizon/resources/Technical-manuals/edit-r-lentiviral-sgrna-manual.pdf). Cells were incubated at 37°C for 6 h with 250 μl of serum-free media containing CRISPRi lentivirus at a multiplicity of infection (MOI) of 0.3. Afterwards, 750 μl complete media was added on top of the cells. After 48 h, cells were collected and a serial dilution was performed. Clones were

grown for 2 weeks in the presence of 15 μg/ml blasticidin. Twelve clones were isolated with Scienceware cloning discs (Z374431-100EA) and allowed to grow. Clone 8 was picked for future assays based on the efficiency of sgRNA-mediated protein down-regulation.

### 3D spheroids

To generate MDA-MB-231 cells stably expressing GFP, $8 \times 10^5$ cells/well were seeded into a 6-well plate in 2 μg/ml of pSBtet-GB GFP Luciferase plasmid of complete media without antibiotics. Confluent cells were transfected with 2.5 and 0.25 μg of the sleeping beauty transposon plasmid, pCMV(CAT)T7-SB100, and 250 μl of OptiMEM, 5 μl p3000, and 3.75 μl Lipofectamine 3000, together with both plasmids was added on top of the 2 ml. Media was changed after 6 h. After 48 h, cells were selected with 3 μg/ml blasticidin for 2 weeks and FACS sorted.

3D spheroids were generated by the hanging drop method, and 500 cells (for proliferation assay) or 1,000 cells (for 3D uptake) per 20 μl drop containing 4.8 mg/ml methylcellulose (Sigma-Aldrich) and 20 μg/ml soluble collagen I (BioEngineering) were pipetted on the lid of tissue culture dishes. Lids were turned and put on top of the bottom reservoir of the dish, which was filled with PBS to prevent evaporation. After 48 h, spheroids were embedded in 40 μl of 3 mg/ml rat tail collagen I (Corning) and 3 mg/ml geltrex (Gibco). For 3D uptake assays, 1/5 of the matrix solution was labelled with 20 μg/ml pHrodo containing 0.1 M NaHCO$_3$. Spheroids were incubated at 37°C for 20 min and then media was added into the dishes. For 3D uptake assays, media was changed to 10% DFBS DMEM and 10% DFBS amino acid-free DMEM after 44 h of embedding. Cells were imaged live every 24 h until 72 h post-embedding. 3D uptake was assessed by quantifying the pH-rodo-ECM integrated intensity per spheroid, data was normalised to day 0.

For 3D proliferation assay, media was changed to 10% DFBS DMEM and 10% DFBS amino acid-free DMEM the day after embedding. For drug treatments, 40 μm Nitisinone and the corresponding vehicle DMSO was added. On day 5 post-embedding, spheroids were incubated with 10 μm EdU for 1 day at 37°C and 5% CO$_2$. At day 6, spheroids were fixed by adding 4% PFA containing 10 μg/ml Hoechst 33342 for 20 min at 37°C followed by 2 PBS washes. They were permeabilized for 2 h at RT by IF wash containing 0.2% Triton-X 100, 0.04% Tween-20, and 0.1% BSA in PBS followed by 2 PBS washes. Spheroids were incubated with EdU detection cocktail (Invitrogen, Click-iT EdU Alexa Fluor 555) overnight at 4°C. They were washed twice with PBS and were kept in PBS for imaging with a Nikon A1 confocal microscope.

For invasion assays, media was changed to 10% DFBS amino acid-free DMEM the day after embedding and spheroids were treated with 40 μm Nitisinone, 3 μm FRAX597, or DMSO control. Spheroids were generated from cells transfected with siRNA targeting PAK1, HPDL or non-targeting control and where embedded in 40 μl of 3 mg/ml rat tail collagen I and 3 mg/ml geltrex in 10% DFBS amino acid-free DMEM. Spheroids were imaged at days 1 to 3 post-embedding with a Nikon A1 confocal microscope. Invasion area was calculated by subtracting the core area from the total area.

### Cell migration

CAF-CDMs were generated in 12-well plates as described in [14], and $0.5 \times 10^5$ MDA-MB-231 cells/well were seeded for 4 to 6 h in amino acid-free media, treated as indicated 1 h before imaging by time lapse phase contrast microscopy (Nikon Ti. eclipse with Oko-lab environmental control chamber; Plan Apo 10×, NA 0.45 objective) in an atmosphere of 5% CO$_2$ at 37°C for 16 h, as described in [43]. The length of pseudopods in the direction of cell migration was measured with Image J.

### Survival analysis

The survival analysis was performed using Kaplan–Meier plotter (https://kmplot.com/analysis/) [66], which can assess the effect of genes of interest on survival in 21 cancer types, including breast cancer. Sources for the databases include Gene Expression Omnibus (GEO), European Genome-Phenome Archive (EGA), and The Cancer Genome Atlas (TCGA).

### Statistical analysis

Graphs were created by GraphPad Prism software (version 9 and 10) as super-plots [67], where single data points from individual experiments are presented in the same shade (grey, maroon, and blue) and the mean is represented by black dots. To compare more than 2 data set, one-way ANOVA (Kruskal–Wallis, Dunn's multiple comparisons test) was used when there was 1 independent variable. Two-way ANOVA (Tukey's multiple comparisons test) was performed when there were 2 independent variables. Statistical analysis for non-targeted metabolic profiling was performed by Perseus software and used Student $t$ test (SO = 0.1 and false discovery rate (FDR) = 0.05).

## Supporting information

**S1 Fig. Matrigel supports breast cancer cell growth under starvation.** (A) Schematic, cell proliferation experiments. (B–D) MDA-MB-231 cells were seeded on plastic or 3 mg/ml Matrigel (matr) for 8 days under complete media, glutamine (Gln) or amino acid (AA) starvation, fixed, stained with DRAQ5 and imaged with a Licor Odyssey system. Signal intensity was calculated by Image Studio Lite software. Values are mean ± SEM from 3 independent experiments (the black dots in the bar graph represent the mean of individual experiments). $^*p < 0.05$, $^{**}p < 0.001$, $^{****}p < 0.0001$. (C) Two-way ANOVA, Tukey's multiple comparisons test; (D) Kruskal–Wallis, Dunn's multiple comparisons test. All the raw data associated with this figure are available in S9 Data.
(TIF)

**S2 Fig. The ability to use Matrigel to support cell growth is acquired during breast carcinoma progression.** MCF10A (A–D), MCF10DCIS (E–H) and MCF10CA1 (I–L) cells were seeded on plastic or 3 mg/ml Matrigel for 6 or 8 days under complete media, glutamine (Gln) or amino acid (AA) starvation, fixed, stained with DRAQ5 and imaged with a Licor Odyssey system. Signal intensity was calculated by Image Studio Lite software. Values are mean ± SEM from 3 independent experiments (the black dots in the bar graphs represent the mean of individual experiments). $^*p < 0.05$, $^{**}p < 0.01$, $^{****}p < 0.0001$ (B, C, G, K) two-way ANOVA, Tukey's multiple comparisons test. (H, L) Kruskal–Wallis, Dunn's multiple comparisons test. All the raw data associated with this figure are available in S10 Data.
(TIF)

**S3 Fig. The ECM did not affect cell death under starvation.** (A, F) Schematic, caspase3/7 activation and propidium iodide (PI) staining. MDA-MB-231 cells were seeded on plastic, 2 mg/ml collagen I (coll I) or 3 mg/ml Matrigel (Matr) for (B, G) 3 or (C, H) 8 days in complete media (Com) or amino acid-free media (AA). MCF10CA1 cells were seeded on plastic, 2 mg/ml collagen I (coll I) or 3 mg/ml Matrigel (Matr) for (D, I) 3 or (E, J) 6 days in complete media (Com) or amino acid-free media (AA). Cells were fixed and stained for activated caspase-3/7 (CAS-3/7, B–E) or PI (G–J). Images were collected by ImageXpress micro and analysed by CME software. Values are mean ± SEM of at least 6 replicates from at least 2 independent experiments (the black dots represent the mean of individual experiments). $^*p < 0.05$, $^{**}p < 0.01$, $^{***}p < 0.001$, $^{****}p < 0.0001$ Kruskal–Wallis, Dunn's multiple comparisons test.

All the raw data associated with this figure are available in S11 Data.
(TIF)

**S4 Fig. mTOR activation did not promote cell growth in MCF10A and MCF10A-DCIS cells under starvation.** (A) MCF10A cells were amino acid starved and treated with 5 μm MHY1485 (+) or vehicle control (-) for 24 h, lysed and p-4EBP1 and GAPDH levels were assessed by western blotting. Values are mean ± SEM from 4 independent experiments; *$p < 0.05$ Mann–Whitney test. (B) MCF10A (10A) and MCF10A-DCIS (DCIS) cells were grown on 2 mg/ml collagen I for 4 days under amino acid (AA) starvation, in the presence of 5 μm MHY1485 (MHY) or vehicle control (Ctrl). Cells were fixed, stained with Hoechst 33342, imaged by ImageXpress micro and analysed by MetaXpress software. Values are mean ± SEM from 3 independent experiments (the black dots represent the mean of individual experiments). All the raw data associated with this figure are available in S12 Data.
(TIF)

**S5 Fig. ECM uptake, and not extracellular proteolysis, was required for cell growth under starvation.** MDA-MB-231 cells were plated under complete (Com) or amino acid depleted (AA) media on NHS-fluorescein labelled (A) 2 mg/ml collagen I (coll I) or (B) 3 mg/ml Matrigel-coated dishes for 3 days, in the presence of the lysosomal inhibitor E64d (20 μm) or DMSO (control). (C–E) 2 mg/ml collagen I (coll I) or 3 mg/ml Matrigel-coated dishes were labelled with NHS-Fluorescein (green) and treated with 10% glutaraldehyde for 30 min or left untreated. MDA-MB-231 (C, D) or MCF10CA1 (E) cells were plated on the cross-liked (x-linked) matrices for 3 days under amino acid (AA) starvation in the presence of E64d (20 μm). Cells were fixed and stained for actin (red) and nuclei (blue). Samples were imaged with a Nikon A1 confocal microscope and ECM uptake index was calculated with image J. Bar, 30 μm (C) and 20 μm (E). More than 150 cells per condition in 2 or 3 independent experiments were analysed, the black dots represent the mean of individual experiments. **$p < 0.01$, ****$p < 0.0001$ Kruskal–Wallis, Dunn's multiple comparisons test. MDA-MB-231 (F, G) and MCF10CA1 (H) cells were seeded on untreated (control) or cross-linked (X-linked) 2 mg/ml collagen I (coll I) or 3 mg/ml Matrigel under complete media (com, (F) or amino acid (AA) starvation (G, H) for 6 or 8 days). Cells were fixed, stained with Hoechst 33342, imaged by ImageXpress micro and analysed by MetaXpress software. *$p < 0.05$, ***$p < 0.001$, ****$p < 0.0001$ two-way ANOVA, Tukey's multiple comparisons (G); Kruskal–Wallis, Dunn's multiple comparisons test (H). (I) Schematic, MMP activity. (J) MDA-MB-231 cells were plated in complete media on NHS-fluorescein labelled 2 mg/ml collagen I (coll I) or 3 mg/ml Matrigel-coated dishes for 3 days, in the presence of 20 μm E64d and 10 μm GM6001 (MMPi) or DMSO (control). Cells were imaged and quantified as in (C). Bar, 10 μm. About 150 cells per condition from 3 independent experiments were analysed, the black dots represent the mean of individual experiments. ****$p < 0.0001$ Kruskal–Wallis, Dunn's multiple comparisons test. (K–M) MDA-MB-231 were seeded on (K) plastic or (L, M) 2 mg/ml collagen I under complete (Com) or AA depleted (AA) media for 4 days; 10 μm GM6001 (MMPi) or DMSO (control) were added to cells every 2 days. Cells were fixed and imaged as in (F). Values are mean ± SEM from 3 independent experiments. All the raw data associated with this figure are available in S13 Data.
(TIF)

**S6 Fig. Caveolin 1/2, Dynamin 2/3, and PAK1 controlled ECM internalisation.** MDA-MB-231 cells were transfected with siRNA targeting the indicated genes, seeded on pH-rodo labelled 0.5 mg/ml Matrigel (A) or NF-CDM (B) for 6 h, stained with Hoechst 33342, imaged live with an Opera Phenix microscope, and analysed with Columbus software (A) or imaged

live with a Nikon A1 confocal microscope and quantified with Image J. Scale bar, 20 μm (B). (C) MDA-MB-231, MDA-MB-231 CRISPRi, and MCF10CA1 cells were transfected with an siRNA targeting PAK1 (PAK1-siRNA), a non-targeting siRNA control (nt-siRNA), a synthetic guide RNA targeting PAK1 (PAK1-sgRNA) or a non-targeting synthetic guide RNA control (nt-sgRNA), lysed and PAK1 and GAPDH expression were measured by western blotting. (D) MDA-MB-231 cells were transfected with an siRNA targeting PAK1 (PAK1-siRNA) or a non-targeting siRNA control (nt-siRNA), plated on pH-rodo labelled NF-CDM (red) or Alexa Fluor 555 labelled 1 mg/ml collagen I (coll I, red) for 6 h, stained with Hoechst 33342 (blue) and imaged live with a Nikon A1 confocal microscope or fixed and stained for actin (green) and nuclei (blue). Scale bar, 20 μm. ECM uptake index was quantified with Image J. (E) MCF10A (10A), MCF10A-DCIS (DCIS), and MCF10CA1 (CA1) were lysed and PAK1 and GAPDH expression were measured by western blotting. MDA-MB-231 (F, G) and MDA-MB-231 CRISPRi (G) cells were plated on 2 mg/ml collagen I (coll I, F) or CAF-CDM (G), transfected with an siRNA targeting PAK1 (PAK1-siRNA), a non-targeting siRNA control (nt-siRNA, F, G), a synthetic guide RNA targeting PAK1 (PAK1-sgRNA) and a non-targeting synthetic guide RNA control (nt-sgRNA, G) and cultured in complete media for 6 days. Cells were fixed and stained with Hoechst 33342. MDA-MB-231 cells (H) were grown on CAF-CDM in complete media for 4 days in the presence of 3 μm FRAX597, fixed and stained with Hoechst 33342. Images were collected by ImageXpress micro and analysed by MetaXpress software. Values are mean ± SEM from at least 3 independent experiments. $^*p < 0.05$, $^{**}p < 0.01$, $^{***}p < 0.001$, $^{****}p < 0.0001$ Kruskal–Wallis, Dunn's multiple comparisons test (B, G); Mann–Whitney test (C, D, H). All the raw data associated with this figure are available in S14 Data.
(TIF)

**S7 Fig. Substrate rigidity did not affect ECM uptake and ECM-dependent cell growth under starvation.** (A) 1 mg/ml collagen I gels were treated with 200 mM ribose for 48 h or left untreated and labelled with pH-rodo. MDA-MB-231 cells were seeded for 6 h, stained with Hoechst 33342, imaged live with a Nikon A1 confocal microscope and analysed with Image J. Bar, 20 μm. Values are mean ± SEM from 3 independent experiments. (B) MDA-MB-231 cells were plated on 200 mM ribose-treated or control collagen I, cultured in complete (Com) or amino acid-free (AA) media for 6 days, fixed and stained with Hoechst 33342. Images were collected by ImageXpress micro and analysed by MetaXpress software. Values are mean ± SEM from 3 independent experiments. (C) MDA-MB-231 cells were plated on poly-acrylamide hydrogels of the indicated stiffness coated with 1 mg/ml collagen I, cultured in complete (Com) or amino acid-free (AA) media for 5 days, incubated with EdU for 1 day, fixed and stained with Hoechst 33342 and Click iT EdU imaging kit. The percentage of EdU positive cells was measured with Image J. Values are mean ± SEM from 3 independent experiments. $^*p = 0.0427$, $^{**}p = 0.0044$, Kruskal–Wallis, Dunn's multiple comparisons test. Parts of the figure were drawn by using pictures from Servier Medical Art. Servier Medical Art by Servier is licensed under a Creative Commons Attribution 3.0 Unported License (https://creativecommons.org/licenses/by/3.0/). All the raw data associated with this figure are available in S15 Data.
(TIF)

**S8 Fig. Tyrosine catabolism did not affect cell growth on plastic.** MDA-MB-231 (A–E) and MCF10CA1 (L, M) cells were plated on plastic, 2 mg/ml collagen I or 3 mg/ml Matrigel for 6 days in amino acid-free media. Metabolites were extracted and quantified by non-targeted mass spectrometry. Volcano plots (B, D, L) and enriched metabolic pathways (C, E, M) are presented. These datasets are deposited in ORDA (https://doi.org/10.15131/shef.data.

21608490). (F–I) MDA-MB-231 or MCF10CA1 (Q) cells were plated on CAF-CDM (F), 2 mg/ml collagen I (coll I, G, H, Q) or plastic (I), transfected with siRNA targeting HPD (HPD siRNA), HPDL (HPDL siRNA) or non-targeting siRNA control (Nt siRNA) and cultured in complete, 25% Plasmax or amino acid-free media for 6 days. Cells were fixed and stained with Hoechst 33342. Images were collected by ImageXpress micro and analysed by MetaXpress software. Values are mean ± SEM and from 3 independent experiments. $^*p < 0.05$, $^{**}p < 0.01$, $^{****}p < 0.0001$ Kruskal–Wallis, Dunn's multiple comparisons test. (J) MDA-MB-231 cells were plated on 2 mg/ml collagen I, transfected with siRNA targeting HPDL (HPDL siRNA) or non-targeting siRNA control (Nt siRNA) and cultured in amino acid depleted media for 3 days. Metabolites were extracted and fumarate was measured by targeted mass spectrometry. $^*p = 0.0126$ Mann–Whitney test. (K) MDA-MB-231 cells were grown in amino acid-free media in the presence of C13 Tyrosine for 6 days. Metabolites were extracted and fumarate isotopologue abundance was quantified by targeted mass spectrometry. (N) Metabolites were prepared as in (A) and the levels of phenylalanine, tyrosine, and fumarate were measured by targeted mass spectrometry. (O) MCF10CA1 cells transfected with siRNA targeting HPDL (HPDL siRNA) or non-targeting siRNA control (Nt siRNA), fixed and stained for HPDL (green), actin (red) and nuclei (blue). Images were collected with a Nikon A1 confocal microscope. Bar, 20 μm. HPDL intensity was quantified with ImageJ. (P) MDA-MB-231 cells transfected with siRNA targeting HPD (HPD siRNA) or non-targeting siRNA control (Nt siRNA), the mRNA was extracted and HPD expression was measured by qPCR. Data are mean ± SEM from 3 independent experiments; $^{****}p < 0.0001$, Mann–Whitney test. The numerical data associated with this figure are available in S16 Data.
(TIF)

**S9 Fig. Matrigel supported the growth of MCF10A cells under serum starvation.** MCF10A cells were seeded on plastic or 3 mg/ml Matrigel for 8 days under serum (FBS) starvation, fixed, stained with DRAQ5 and imaged with a Licor Odyssey system. Signal intensity was calculated by Image Studio Lite software. Values are mean ± SEM from 3 independent experiments. $^*p = 0.0230$ two-way ANOVA, Tukey's multiple comparisons test. All the raw data associated with this figure are available in S17 Data.
(TIF)

**S10 Fig. TCA intermediates were up-regulated on collagen I under amino acid starvation.** MDA-MB-231 cells were plated on plastic or 2 mg/ml collagen I for 6 days in amino acid-free media. Metabolites were extracted and quantified by non-targeted mass spectrometry. $^{****}p < 0.0001$ two-way ANOVA, Tukey's multiple comparisons test. All the raw data associated with this figure are available in S18 Data.
(TIF)

**S11 Fig. HPDL expression was up-regulated in basal-like breast cancers.** HPDL expression in human tumours and normal tissues, stratified by breast cancer subtypes. Data were obtained from GEPIA 2 (http://gepia2.cancer-pku.cn), $^*p < 0.01$.
(TIF)

**S1 Raw Image. Original western blot image used to prepare Fig 5B.**
(TIF)

**S2 Raw Image. Original western blot image used to prepare Fig 4A.**
(TIF)

**S3 Raw Image. Original western blot image used to prepare Fig 6C.**
(TIF)

**S4 Raw Image. Original western blot image used to prepare Fig 6E.**
(TIF)

**S1 Data. Numerical data used for the generation of the graphs presented in Fig 1.** The biological replicates are colour coded.
(XLSX)

**S2 Data. Numerical data used for the generation of the graphs presented in Fig 2.** The biological replicates are colour coded.
(XLSX)

**S3 Data. Numerical data used for the generation of the graphs presented in Fig 3.** The biological replicates are colour coded.
(XLSX)

**S4 Data. Numerical data used for the generation of the graphs presented in Fig 4.** The biological replicates are colour coded.
(XLSX)

**S5 Data. Numerical data used for the generation of the graphs presented in Fig 5.** The biological replicates are colour coded.
(XLSX)

**S6 Data. Numerical data used for the generation of the graphs presented in Fig 6.** The biological replicates are colour coded.
(XLSX)

**S7 Data. Numerical data used for the generation of the graphs presented in Fig 7.** The biological replicates are colour coded.
(XLSX)

**S8 Data. Numerical data used for the generation of the graphs presented in Fig 8.** The biological replicates are colour coded.
(XLSX)

**S9 Data. Numerical data used for the generation of the graphs presented in S1 Fig.** The biological replicates are colour coded.
(XLSX)

**S10 Data. Numerical data used for the generation of the graphs presented in S2 Fig.** The biological replicates are colour coded.
(XLSX)

**S11 Data. Numerical data used for the generation of the graphs presented in S3 Fig.** The biological replicates are colour coded.
(XLSX)

**S12 Data. Numerical data used for the generation of the graphs presented in S4 Fig.** The biological replicates are colour coded.
(XLSX)

**S13 Data. Numerical data used for the generation of the graphs presented in S5 Fig.** The biological replicates are colour coded.
(XLSX)

**S14 Data. Numerical data used for the generation of the graphs presented in S6 Fig.** The biological replicates are colour coded.
(XLSX)

**S15 Data. Numerical data used for the generation of the graphs presented in S7 Fig.** The biological replicates are colour coded.
(XLSX)

**S16 Data. Numerical data used for the generation of the graphs presented in S8 Fig.** The biological replicates are colour coded.
(XLSX)

**S17 Data. Numerical data used for the generation of the graphs presented in S9 Fig.** The biological replicates are colour coded.
(XLSX)

**S18 Data. Numerical data used for the generation of the graphs presented in S10 Fig.** The biological replicates are colour coded.
(XLSX)

## Acknowledgments

Metabolomics was performed at the Faculty of Science Biological Mass Spectrometry Facility, University of Sheffield (biOMICS). Proteomics was performed at the Faculty of Science Biological Mass Spectrometry Facility, University of Sheffield (biOMICS), funded by the MRC (grant MR/X012220/1). Imaging work was performed at the Wolfson Light Microscopy Facility, University of Sheffield, using the Nikon A1 confocal/TIRF microscope. High-throughput imaging was performed in the Sheffield RNAi Screening Facility and in the RNAi screening facility in IMCB, Singapore. qPCR analysis was performed in collaboration with the Tsakiridis lab at the University of Sheffield. We would like to acknowledge Marga Albu for developing the Image J macro used for ECM uptake quantification and the "ECM and stroma" group at the University of Sheffield for providing valuable feedback which helped shaping this study. The Wolfson Light Microscopy Facility, University of Sheffield, is funded by the Wellcome Trust (grant WT093134AIA).

## Author Contributions

**Conceptualization:** Mona Nazemi, Montserrat Llanses Martinez, Mark O. Collins, Elena Rainero.

**Formal analysis:** Mona Nazemi, Bian Yanes, Montserrat Llanses Martinez, Khoa Pham.

**Funding acquisition:** Mark O. Collins, Frederic Bard, Elena Rainero.

**Investigation:** Mona Nazemi, Bian Yanes, Montserrat Llanses Martinez, Heather J. Walker, Khoa Pham, Elena Rainero.

**Methodology:** Mona Nazemi, Montserrat Llanses Martinez.

**Supervision:** Mark O. Collins, Frederic Bard, Elena Rainero.

**Writing – original draft:** Mona Nazemi, Elena Rainero.

**Writing – review & editing:** Elena Rainero.

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
