## [Editor Report · Decision Letter 0]

31 Mar 2023

Dear Dr Rainero, 

Thank you for submitting your manuscript from Review Commons entitled "The extracellular matrix supports breast cancer cell growth under amino acid starvation by promoting tyrosine catabolism" for consideration as a Research Article by PLOS Biology.

Your manuscript has now been evaluated by the PLOS Biology editorial staff, as well as by an academic editor with relevant expertise, and I am writing to let you know that we would like to invite you to submit a revised version in response to the reports at Review Commons as detailed in your revision plan.

However, before we can invite a revision, we need you to complete your submission by providing the metadata that is required for full assessment. To this end, please login to Editorial Manager where you will find the paper in the 'Submissions Needing Revisions' folder on your homepage. Please click 'Revise Submission' from the Action Links and complete all additional questions in the submission questionnaire.

Once your full submission is complete, your paper will undergo a series of checks in preparation for peer review. After your manuscript has passed the checks it will be sent out for review. To provide the metadata for your submission, please Login to Editorial Manager (https://www.editorialmanager.com/pbiology) within two working days, i.e. by Apr 02 2023 11:59PM.

Kind regards,

Richard

Richard Hodge, PhD

Associate Editor, PLOS Biology

rhodge@plos.org

PLOS

---

## [Editor Report · Decision Letter 1]

6 Apr 2023

Dear Dr Rainero,

Thank you very much for submitting your manuscript "The extracellular matrix supports breast cancer cell growth under amino acid starvation by promoting tyrosine catabolism" for consideration as a Research Article at PLOS Biology. As you know, your manuscript and plan of revision have been evaluated by the PLOS Biology editors and by an Academic Editor with relevant expertise.

Based on your responses to the reviews from Reviews Commons, we would welcome re-submission of a revised version that takes into account the reviewers' comments. After discussions with the academic editor, we do agree with Reviewer #2 that the addition of some in vivo data would strengthen the findings, although this could be limited to just including the breast tumor xenograft model.

We cannot make any decision about publication until we have seen the revised manuscript and your response to the reviewers' comments at Review Commons. Your revised manuscript is also likely to be sent for further evaluation by the original reviewers.

We expect to receive your revised manuscript within 3 months. Please email us (plosbiology@plos.org) if you have any questions or concerns, or would like to request an extension. At this stage, your manuscript remains formally under active consideration at our journal; please notify us by email if you do not intend to submit a revision so that we may withdraw it.

**IMPORTANT - SUBMITTING YOUR REVISION**

*Re-submission Checklist*

*Published Peer Review*

*PLOS Data Policy*

*Blot and Gel Data Policy*

Sincerely,

Richard

Richard Hodge, PhD

Associate Editor, PLOS Biology

rhodge@plos.org

PLOS

---

## [Decision Letter · Decision Letter 2]

13 Oct 2023

Dear Dr Rainero,

Thank you for your patience while we considered your revised manuscript "The extracellular matrix supports breast cancer cell growth under amino acid starvation by promoting tyrosine catabolism" for publication as a Research Article at PLOS Biology. This revised version of your manuscript has been evaluated by the PLOS Biology editors, the Academic Editor and the original reviewers at Review Commons.

Based on the reviews, I am pleased to say that we are likely to accept this manuscript for publication, provided you satisfactorily address the remaining points raised by the reviewers by discussing of the limitations of the study in regards to the in vivo validation and contextualizing the findings amongst the different breast cancer subtypes. In addition, please also make sure to address the following data and other policy-related requests that I have provided below (A-D):

(A) You may be aware of the PLOS Data Policy, which requires that all data be made available without restriction: http://journals.plos.org/plosbiology/s/data-availability. For more information, please also see this editorial: http://dx.doi.org/10.1371/journal.pbio.1001797

-Supplementary files (e.g., excel). Please ensure that all data files are uploaded as 'Supporting Information' and are invariably referred to (in the manuscript, figure legends, and the Description field when uploading your files) using the following format verbatim: S1 Data, S2 Data, etc. Multiple panels of a single or even several figures can be included as multiple sheets in one excel file that is saved using exactly the following convention: S1_Data.xlsx (using an underscore).

-Deposition in a publicly available repository. Please also provide the accession code or a reviewer link so that we may view your data before publication. 

Figure 1B-G, 1I-K, 2B-D, 2F-H, 2J-N, 3B-C, 3E-F, 4B, 4D, 4F, 4H-L, 5C-J, 5L-M, 6B-D, 6H-R, 7B-C, 8B-C, 8E-F, S1B-D, S2B-D, S2F-H, S2J-L, S3B-E, S3G-J, S4A-B, S5A-B, S5D-H, S5J-M, S6A-H, S7A-C, S8B-K, S8N, S6P-Q, S9, S10

(B) For the mass spectrometry data, we suggest depositing the datasets in the ProteomeXchange repository (https://www.proteomexchange.org/) due to the potentially large size of the files.

(C) Please also ensure that each of the relevant figure legends in your manuscript include information on *WHERE THE UNDERLYING DATA CAN BE FOUND*, and ensure your supplemental data file/s has a legend.

(D) Please ensure that your Data Statement in the submission system accurately describes where your data can be found and is in final format, as it will be published as written there.

We expect to receive your revised manuscript within two weeks. 

*Published Peer Review History*

*Press*

Kind regards,

Richard

Richard Hodge, PhD

rhodge@plos.org

Reviewer remarks:

Reviewer #1: The revised form of the manuscript now submitted from the authors is more clear and complete compared the first version. I appreciate the work done from the authors to answer to the doubts and questions raised during the last revision.

Certainly, the presence of data on the involvement of ECM in supporting aggressive breast cancer growth in vivo would have definitely reinforced the conclusions of the study and the lack of an in vivo validation of the results obtained in vitro could represent a weakness of the manuscript, but I think that the results are new and well presented. The data here shown obtained with a good experimental design can contribute to expand the knowledges on the factors that promote cancer metabolic plasticity, which is exploited form cancer cell to obtain advantages in term of growth, survival and progression.

In conclusion, the manuscript submitted by Nazemi et al. is eligible for the publication, but it' is necessary increase the quality of the images of the Figures, unless that their low quality is dependent only on submitted file. 

Reviewer #2: The authors have now answered all our queries and we think this exciting mansucript is ready for publication. One very minor questions, in the rebuttal the authors state "- Q:The PAK1 expression level blots in the knockdown experiments should be quantified from N=3. A: We have not included the quantification of the western blots in figure S5C." I am sure this is a typo, because there are no western blot images in FigS5?

Reviewer #3: I sincerely thank the author for taking into account all of my comments. I have now red the revised version of hte manuscript and while, I found that the lack of in vivo data is harmful for the publication of the manuscript, I have no other concern regarding the experiemental data sets that are presented in the revised version.

---

## [Editor Report · Decision Letter 3]

26 Oct 2023

Dear Dr Rainero,

Thank you for the submission of your revised Research Article "The extracellular matrix supports breast cancer cell growth under amino acid starvation by promoting tyrosine catabolism" for publication in PLOS Biology. On behalf of my colleagues and the Academic Editor, Heather Christofk, I am pleased to say that we can accept your manuscript for publication, provided you address any remaining formatting and reporting issues. These will be detailed in an email you should receive within 2-3 business days from our colleagues in the journal operations team; no action is required from you until then. Please note that we will not be able to formally accept your manuscript and schedule it for publication until you have completed any requested changes.

PRESS

Best wishes, 

Richard

Richard Hodge, PhD

rhodge@plos.org

PLOS
